# Deep Operator Learning Lessens the Curse of Dimensionality for PDEs

**Ke Chen**                                                          *kechen@umd.edu*
*Department of Mathematics*
*University of Maryland, College Park*

**Chunmei Wang**                                              *chunmei.wang@ufl.edu*
*Department of Mathematics*
*University of Florida*

**Haizhao Yang**                                                     *hzyang@umd.edu*
*Department of Mathematics*
*University of Maryland, College Park*

**Reviewed on OpenReview:** *https://openreview.net/forum?id=zmBFzuT2DN*

## Abstract

Deep neural networks (DNNs) have achieved remarkable success in numerous domains, and their application to PDE-related problems has been rapidly advancing. This paper provides an estimate for the generalization error of learning Lipschitz operators over Banach spaces using DNNs with applications to various PDE solution operators. The goal is to specify DNN width, depth, and the number of training samples needed to guarantee a certain testing error. Under mild assumptions on data distributions or operator structures, our analysis shows that deep operator learning can have a relaxed dependence on the discretization resolution of PDEs and, hence, lessen the curse of dimensionality in many PDE-related problems including elliptic equations, parabolic equations, and Burgers equations. Our results are also applied to give insights about discretization-invariance in operator learning.

## 1 Introduction

Nonlinear operator learning aims to learn a mapping from a parametric function space to the solution space of specific partial differential equation (PDE) problems. It has gained significant importance in various fields, including order reduction Peherstorfer & Willcox (2016), parametric PDEs Lu et al. (2021b); Li et al. (2021), inverse problems Khoo & Ying (2019), and imaging problems Deng et al. (2020); Qiao et al. (2021); Tian et al. (2020). Deep neural networks (DNNs) have emerged as state-of-the-art models in numerous machine learning tasks Graves et al. (2013); Miotto et al. (2018); Krizhevsky et al. (2017), attracting attention for their applications to engineering problems where PDEs have long been the dominant model. Consequently, deep operator learning has emerged as a powerful tool for nonlinear PDE operator learning Lanthaler et al. (2022); Li et al. (2021); Nelsen & Stuart (2021); Khoo & Ying (2019). The typical approach involves discretizing the computational domain and representing functions as vectors that tabulate function values on the discretization mesh. A DNN is then employed to learn the map between finite-dimensional spaces. While this method has been successful in various applications Lin et al. (2021); Cai et al. (2021), its computational cost is high due to its dependence on the mesh. This implies that retraining of the DNN is necessary when using a different domain discretization. To address this issue, Li et al. (2021); Lu et al. (2022); Ong et al. (2022) have been proposed for problems with sparsity structures and discretization-invariance properties. Another line of works for learning PDE operators are generative models, including Generative adversarial models (GANs) and its variants Rahman et al. (2022); Botelho et al. (2020); Kadeethum et al. (2021) and diffusion models Wang et al. (2023). These methods can deal with discontinuous features, whereas neural

network based methods are mainly applied to operators with continuous input and output. However, most of generative models for PDE operator learning are limited to empirical study and theoretical foundations are in lack.

Despite the empirical success of deep operator learning in numerous applications, its statistical learning theory is still limited, particularly when dealing with infinite-dimensional ambient spaces. The learning theory generally comprises three components: approximation theory, optimization theory, and generalization theory. Approximation theory quantifies the expressibility of various DNNs as surrogates for a class of operators. The universal approximation theory for certain classes of functions Cybenko (1989); Hornik (1991) forms the basis of the approximation theory for DNNs. It has been extended to other function classes, such as continuous functions Shen et al. (2019); Yarotsky (2021); Shen et al. (2021), certain smooth functions Yarotsky (2018); Lu et al. (2021a); Suzuki (2018); Adcock et al. (2022), and functions with integral representations Barron (1993); Ma et al. (2022). However, compared to the abundance of theoretical works on approximation theory for high-dimensional functions, the approximation theory for operators, especially between infinite-dimensional spaces, is quite limited. Seminal quantitative results have been presented in Kovachki et al. (2021); Lanthaler et al. (2022).

In contrast to approximation theory, generalization theory aims to address the following question:

*How many training samples are required to achieve a certain testing error?*

This question has been addressed by numerous statistical learning theory works for function regression using neural network structures Bauer & Kohler (2019); Chen et al. (2022); Farrell et al. (2021); Kohler & Krzyżak (2005); Liu et al. (2021); Nakada & Imaizumi (2020); Schmidt-Hieber (2020). In a $d$-dimensional learning problem, the typical error decay rate is on the order of $n^{-\mathcal{O}(1/d)}$ as the number of samples $n$ increases. The fact that the exponent is very small for large dimensionality $d$ is known as the *curse of dimensionality* (CoD) Stone (1982). Recent studies have demonstrated that DNNs can achieve faster decay rates when dealing with target functions or function domains that possess low-dimensional structures Chen et al. (2019; 2022); Cloninger & Klock (2020); Nakada & Imaizumi (2020); Schmidt-Hieber (2019); Shen et al. (2019). In such cases, the decay rate becomes independent of the domain discretization, thereby lessening the CoD Bauer & Kohler (2019); Chkifa et al. (2015); Suzuki (2018). However, it is worth noting that most existing works primarily focus on functions between finite-dimensional spaces. To the best of our knowledge, previous results de Hoop et al. (2021); Lanthaler et al. (2022); Lu et al. (2021b); Liu et al. (2022) provide the only generalization analysis for infinite-dimensional functions. Our work extends the findings of Liu et al. (2022) by generalizing them to Banach spaces and conducting new analyses within the context of PDE problems. The removal of the inner-product assumption is crucial in our research, enabling us to apply the estimates to various PDE problems where previous results do not apply. This is mainly because the suitable space for functions involved in most practical PDE examples are Banach spaces where the inner-product is not well-defined. Examples include the conductivity media function in the parametric elliptic equation, the drift force field in the transport equation, and the solution to the viscous Burgers equation that models continuum fluid. See more details in Section 3.

## 1.1 Our contributions

The main objective of this study is to investigate the reasons behind the reduction of the CoD in PDE-related problems achieved by deep operator learning. We observe that many PDE operators exhibit a composition structure consisting of linear transformations and element-wise nonlinear transformations with a small number of inputs. DNNs are particularly effective in learning such structures due to their ability to evaluate networks point-wise. We provide an analysis of the approximation and generalization errors and apply it to various PDE problems to determine the extent to which the CoD can be mitigated. Our contributions can be summarized as follows:

- Our work provides a theoretical explanation to why CoD is lessened in PDE operator learning. We extend the generalization theory in Liu et al. (2022) from Hilbert spaces to Banach spaces, and apply it to several PDE examples. Such extension holds great significance as it overcomes a limitation in previous works, which primarily focused on Hilbert spaces and therefore lacked applicability

in machine learning for practical PDEs problems. Comparing to Liu et al. (2022), our estimate circumvented the inner-product structure at the price of a non-decaying noise estimate. This is a tradeoff of accuracy for generalization to Banach space. Our work tackles a broader range of PDE operators that are defined on Banach spaces. In particular, five PDE examples are given in Section 3 whose solution spaces are not Hilbert spaces.

○ Unlike existing works such as Lanthaler et al. (2022), which only offer posterior analysis, we provide an a priori estimate for PDE operator learning. Our estimate does not make any assumptions about the trained neural network and explicitly quantifies the required number of data samples and network sizes based on a given testing error criterion. Furthermore, we identify two key structures—low-dimensional and low-complexity structures (described in assumptions 5 and 6, respectively)—that are commonly present in PDE operators. We demonstrate that both structures exhibit a sample complexity that depends on the essential dimension of the PDE itself, weakly depending on the PDE discretization size. This finding provides insights into why deep operator learning effectively mitigates the CoD.

○ Most operator learning theories consider fixed-size neural networks. However, it is important to account for neural networks with discretization invariance properties, allowing training and evaluation on PDE data of various resolutions. Our theory is flexible and can be applied to derive error estimates for discretization invariant neural networks.

## 1.2 Organization

In Section 2, we introduce the neural network structures and outline the assumptions made on the PDE operator. Furthermore, we present the main results for generic PDE operators, and PDE operators that have low-dimensional structure or low-complexity structure. At the end of the section, we show that the main results are also valid for discretization invariant neural networks. In Section 3, we show that the assumptions are satisfied and provide explicit estimates for five different PDEs. Finally, in Section 4, we discuss the limitations of our current work.

## 2 Problem setup and main results

**Notations.** In a general Banach space $\mathcal{X}$, we represent its associated norm as $\|\cdot\|_{\mathcal{X}}$. Additionally, we denote $E_{\mathcal{X}}^n$ as the encoder mapping from the Banach space $\mathcal{X}$ to a Euclidean space $\mathbb{R}^{d_{\mathcal{X}}}$, where $d_{\mathcal{X}}$ denotes the encoding dimension. Similarly, we denote the decoder for $\mathcal{X}$ as $D_{\mathcal{X}}^n : \mathbb{R}^{d_{\mathcal{X}}} \to \mathcal{X}$. The $\Omega$ notation for neural network parameters in the main results section 2.2 denotes a lower bound estimate, that is, $x = \Omega(y)$ means there exists a constant $C > 0$ such that $x \geq Cy$. The $\mathcal{O}$ notation denotes an upper bound estimate, that is, $x = \mathcal{O}(y)$ means there exists a constant $C > 0$ such that $x \leq Cy$.

## 2.1 Operator learning and loss functions

We consider a general nonlinear PDE operator $\Phi : \mathcal{X} \ni u \mapsto v \in \mathcal{Y}$ over Banach spaces $\mathcal{X}$ and $\mathcal{Y}$. In this context, the input variable $u$ typically represents the initial condition, the boundary condition, or a source of a specific PDE, while the output variable $v$ corresponds to the PDE solution or partial measurements of the solution. Our objective is to train a DNN denoted as $\phi(u; \theta)$ to approximate the target nonlinear operator $\Phi$ using a given data set $\mathcal{S} = \{(u_i, v_i), v_i = \Phi(u_i) + \varepsilon_i, i = 1, \ldots, 2n\}$. The data set $\mathcal{S}$ is divided into $\mathcal{S}_1^n = \{(u_i, v_i), v_i = \Phi(u_i) + \varepsilon_i, i = 1, \ldots, n\}$ that is used to train the encoder and decoders, and a training data set $\mathcal{S}_2^n = \{(u_i, v_i), v_i = \Phi(u_i) + \varepsilon_i, i = n + 1, \ldots, 2n\}$. Both $\mathcal{S}_1^n$ and $\mathcal{S}_2^n$ are generated independently and identically distributed (i.i.d.) from a random measure $\gamma$ over $\mathcal{X}$, with $\varepsilon_i$ representing random noise.

In practical implementations, DNNs operate on finite-dimensional spaces. Therefore, we utilize empirical encoder-decoder pairs, namely $E_{\mathcal{X}}^n : \mathcal{X} \to \mathbb{R}^{d_{\mathcal{X}}}$ and $D_{\mathcal{X}}^n : \mathbb{R}^{d_{\mathcal{X}}} \to \mathcal{X}$, to discretize $u \in \mathcal{X}$. Similarly, we employ empirical encoder-decoder pairs, $E_{\mathcal{Y}}^n : \mathcal{Y} \to \mathbb{R}^{d_{\mathcal{Y}}}$ and $D_{\mathcal{Y}}^n : \mathbb{R}^{d_{\mathcal{Y}}} \to \mathcal{Y}$, for $v \in \mathcal{Y}$. These encoder-decoder pairs are trained using the available data set $\mathcal{S}_1^n$ or manually designed such that $D_{\mathcal{X}}^n \circ E_{\mathcal{X}}^n \approx \mathbb{I}_{d_{\mathcal{X}}}$ and $D_{\mathcal{Y}}^n \circ E_{\mathcal{Y}}^n \approx \mathbb{I}_{d_{\mathcal{Y}}}$. A common example of empirical encoders and decoders is the discretization operator, which

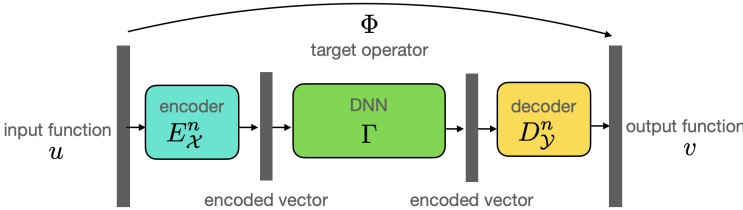

Figure 1: The target nonlinear operator $\Phi : u \mapsto v$ is approximated by compositions of an encoder $E_\mathcal{X}^n$, a DNN function $\Gamma$, and a decoder $D_\mathcal{Y}^n$. The finite dimensional operator $\Gamma$ is learned via the optimization problem equation 1.

maps a function to a vector representing function values at discrete mesh points. Other examples include finite element projections and spectral methods, which map functions to coefficients of corresponding basis functions. Our goal is to approximate the encoded PDE operator using a finite-dimensional operator $\Gamma$ so that $\Phi \approx D_\mathcal{Y}^n \circ \Gamma \circ E_\mathcal{X}^n$. Refer to Figure 1 for an illustration. This approximation is achieved by solving the following optimization problem:

$$\Gamma_{\mathrm{NN}} \in \operatorname*{argmin}_{\Gamma \in \mathcal{F}_{\mathrm{NN}}} \frac{1}{n} \sum_{i=1}^{n} \|\Gamma \circ E_\mathcal{X}^n(u_i) - E_\mathcal{Y}^n(v_i)\|_2^2. \tag{1}$$

Here the function space $\mathcal{F}_{\mathrm{NN}}$ represents a collection of rectified linear unit (ReLU) feedforward DNNs denoted as $f(x)$, which are defined as follows:

$$f(x) = W_L \phi_{L-1} \circ \phi_{L-2} \circ \cdots \circ \phi_1(x) + b_L, \quad \phi_i(x) := \sigma(W_i x + b_i), i = 1, \ldots, L-1, \tag{2}$$

where $\sigma$ is the ReLU activation function $\sigma(x) = \max\{x, 0\}$, and $W_i$ and $b_i$ represent weight matrices and bias vectors, respectively. The ReLU function is evaluated pointwise on all entries of the input vector. In practice, the functional space $\mathcal{F}_{\mathrm{NN}}$ is selected as a compact set comprising all ReLU feedforward DNNs. This work investigates two distinct architectures within $\mathcal{F}_{\mathrm{NN}}$. The first architecture within $\mathcal{F}_{\mathrm{NN}}$ is defined as follows:

$$\mathcal{F}_{\mathrm{NN}}(d, L, p, K, \kappa, M) = \{\Gamma = [f_1, f_2, ..., f_d]^\top : \text{ for each } k = 1, ..., d, f_k(x) \text{ is in the form of (2)}$$

$$\text{with } L \text{ layers, width bounded by } p, \|f_k\|_\infty \le M, \ \|W_l\|_{\infty,\infty} \le \kappa, \|b_l\|_\infty \le \kappa, \ \sum_{l=1}^{L} \|W_l\|_0 + \|b_l\|_0 \le K\}, \tag{3}$$

where $\|f\|_\infty = \max_x |f(x)|$, $\|W\|_{\infty,\infty} = \max_{i,j} |W_{i,j}|$, $\|b\|_\infty = \max_i |b_i|$ for any function $f$, matrix $W$, and vector $b$ with $\|\cdot\|_0$ denoting the number of nonzero elements of its argument. The functions in this architecture satisfy parameter bounds with limited cardinalities. The second architecture relaxes some of the constraints compared to the first architecture; i.e.,

$$\mathcal{F}_{\mathrm{NN}}(d, L, p, M) = \{\Gamma = [f_1, f_2, ..., f_d]^\top : \text{ for each } k = 1, ..., d, f_k(x) \text{ is in the form of (2)}$$

$$\text{with } L \text{ layers, width bounded by } p, \|f_k\|_\infty \le M\}. \tag{4}$$

When there is no ambiguity, we use the notation $\mathcal{F}_{\mathrm{NN}}$ and omit its associated parameters.

We consider the following assumptions on the target PDE map $\Phi$, the encoders $E_\mathcal{X}^n, E_\mathcal{Y}^n$, the decoders $D_\mathcal{X}^n, D_\mathcal{Y}^n$, and the data set $\mathcal{S}$ in our theoretical framework.

**Assumption 1** (Compactly supported measure)**.** The probability measure $\gamma$ is supported on a compact set $\Omega_\mathcal{X} \subset \mathcal{X}$. For any $u \in \Omega_\mathcal{X}$, there exists $R_\mathcal{X} > 0$ such that $\|u\|_\mathcal{X} \le R_\mathcal{X}$. Here, $\|\cdot\|_\mathcal{X}$ denotes the associated norm of the space $\mathcal{X}$.

**Assumption 2** (Lipschitz operator)**.** There exists $L_\Phi > 0$ such that for any $u_1, u_2 \in \Omega_\mathcal{X}$,

$$\|\Phi(u_1) - \Phi(u_2)\|_\mathcal{Y} \le L_\Phi \|u_1 - u_2\|_\mathcal{X}.$$

Here, $\|\cdot\|_\mathcal{Y}$ denotes the associated norm of the space $\mathcal{Y}$.

*Remark* 1. Assumption 1 and Assumption 2 imply that the images $v = \Phi(u)$ are bounded by $R_{\mathcal{Y}} := L_\Phi R_{\mathcal{X}}$ for all $u \in \Omega_{\mathcal{X}}$. The Lipschitz constant $L_\Phi$ will be explicitly computed in Section 3 for different PDE operators.

**Assumption 3** (Lipschitz encoders and decoders). The empirical encoders and decoders $E_{\mathcal{X}}^n, D_{\mathcal{X}}^n, E_{\mathcal{Y}}^n, D_{\mathcal{Y}}^n$ satisfy the following properties:

$$E_{\mathcal{X}}^n(0_{\mathcal{X}}) = \mathbf{0} \,, D_{\mathcal{X}}^n(\mathbf{0}) = 0_{\mathcal{X}} \,, E_{\mathcal{Y}}^n(0_{\mathcal{Y}}) = \mathbf{0} \,, D_{\mathcal{Y}}^n(\mathbf{0}) = 0_{\mathcal{Y}} \,,$$

where $\mathbf{0}$ denotes the zero vector and $0_{\mathcal{X}}, 0_{\mathcal{Y}}$ denote the zero function in $\mathcal{X}$ and $\mathcal{Y}$, respectively. Moreover, we assume all empirical encoders are Lipschitz operators such that

$$\|E_{\mathcal{P}}^n u_1 - E_{\mathcal{P}}^n u_2\|_2 \leq L_{E_{\mathcal{P}}^n}\|u_1 - u_2\|_{\mathcal{P}} \,, \quad \mathcal{P} = \mathcal{X}, \mathcal{Y} \,,$$

where $\|\cdot\|_2$ denotes the Euclidean $L^2$ norm, $\|\cdot\|_{\mathcal{P}}$ denotes the associated norm of the Banach space $\mathcal{P}$, and $L_{E_{\mathcal{P}}^n}$ is the Lipschitz constant of the encoder $E_{\mathcal{P}}^n$. Similarly, we also assume that the decoders $D_{\mathcal{P}}^n, \mathcal{P} = \mathcal{X}, \mathcal{Y}$ are also Lipschitz with constants $L_{D_{\mathcal{P}}^n}$.

**Assumption 4** (Noise). For $i = 1, \ldots, 2n$, the noise $\varepsilon_i$ satisfies

1. $\varepsilon_i$ is independent of $u_i$;

2. $\mathbb{E}[\varepsilon_i] = 0$;

3. There exists $\sigma > 0$ such that $\|\varepsilon_i\|_{\mathcal{Y}} \leq \sigma$.

*Remark* 2. The above assumptions on the noise and Lipschitz encoders imply that $\|E_{\mathcal{Y}}^n(\Phi(u_i) + \varepsilon_i) - E_{\mathcal{Y}}^n(\Phi(u))\|_2 \leq L_{E_{\mathcal{Y}}^n}\sigma$.

## 2.2 Main Results

For a trained neural network $\Gamma_{\mathrm{NN}}$ over the data set $\mathcal{S}$, we denote its generalization error as

$$\mathcal{E}_{gen}(\Gamma_{\mathrm{NN}}) := \mathbb{E}_{\mathcal{S}}\mathbb{E}_{u \sim \gamma}\left[\|D_{\mathcal{Y}}^n \circ \Gamma_{\mathrm{NN}} \circ E_{\mathcal{X}}^n(u) - \Phi(u)\|_{\mathcal{Y}}^2\right] \,.$$

Note that we omit its dependence on $\mathcal{S}$ in the notation. We also define the following quantity,

$$\mathcal{E}_{\mathrm{noise,proj}} := L_\Phi^2 \mathbb{E}_{\mathcal{S}}\mathbb{E}_u\left[\|\Pi_{\mathcal{X},d_{\mathcal{X}}}^n(u) - u\|_{\mathcal{X}}^2\right] + \mathbb{E}_{\mathcal{S}}\mathbb{E}_{w \sim \Phi_\# \gamma}\left[\|\Pi_{\mathcal{Y},d_{\mathcal{Y}}}^n(w) - w\|_{\mathcal{Y}}^2\right] + \sigma^2 + n^{-1} \,,$$

where $\Pi_{\mathcal{X},d_{\mathcal{X}}}^n := D_{\mathcal{X}}^n \circ E_{\mathcal{X}}^n$ and $\Pi_{\mathcal{Y},d_{\mathcal{Y}}}^n := D_{\mathcal{Y}}^n \circ E_{\mathcal{Y}}^n$ denote the encoder-decoder projections on $\mathcal{X}$ and $\mathcal{Y}$ respectively. Here the first term shows that the encoder/decoder projection error $\mathbb{E}_{\mathcal{S}}\mathbb{E}_u\left[\|\Pi_{\mathcal{X},d_{\mathcal{X}}}^n(u) - u\|_{\mathcal{X}}^2\right]$ for $\mathcal{X}$ is amplified by the Lipschitz constant; the second term is the encoder/decoder projection error for $\mathcal{Y}$; the third term stands for the noise; and the last term is a small quantity $n^{-1}$. It will be shown later that this quantity appears frequently in our main results.

**Theorem 1.** *Suppose Assumptions 1-4 hold. Let $\Gamma_{\mathrm{NN}}$ be the minimizer of the optimization problem equation 1, with the network architecture $\mathcal{F}_{\mathrm{NN}}(d_{\mathcal{Y}}, L, p, K, \kappa, M)$ defined in equation 3 with parameters*

$$L = \Omega\left(\ln(\frac{n}{d_{\mathcal{Y}}})\right), \quad p = \Omega\left(d_{\mathcal{Y}}^{\frac{2-d_{\mathcal{X}}}{2+d_{\mathcal{X}}}} n^{\frac{d_{\mathcal{X}}}{2+d_{\mathcal{X}}}}\right),$$
$$K = \Omega(pL), \quad \kappa = \Omega(M^2), \quad M \geq \sqrt{d_{\mathcal{Y}}} L_{E_{\mathcal{Y}}^n} R_{\mathcal{Y}} \,,$$

*where the notation $\Omega$ contains constants that solely depend on $L_{E_{\mathcal{Y}}^n}, L_{D_{\mathcal{Y}}^n}, L_{E_{\mathcal{X}}^n}, L_{D_{\mathcal{X}}^n}, R_{\mathcal{X}}$ and $d_{\mathcal{X}}$. Then there holds*

$$\mathcal{E}_{gen}(\Gamma_{\mathrm{NN}}) \lesssim d_{\mathcal{Y}}^{\frac{6+d_{\mathcal{X}}}{2+d_{\mathcal{X}}}} n^{-\frac{2}{2+d_{\mathcal{X}}}}(1 + L_\Phi^{2-d_{\mathcal{X}}})\left(\ln^3 \frac{n}{d_{\mathcal{Y}}} + \ln^2 n\right) + \mathcal{E}_{noise,proj} \,.$$

*Here $\lesssim$ contains constants that solely depend on $L_{E_{\mathcal{Y}}^n}, L_{D_{\mathcal{Y}}^n}, L_{E_{\mathcal{X}}^n}, L_{D_{\mathcal{X}}^n}, R_{\mathcal{X}}$ and $d_{\mathcal{X}}$.*

**Theorem 2.** *Suppose Assumptions 1-4 hold ture. Let $\Gamma_{\mathrm{NN}}$ be the minimizer of the optimization problem equation 1 with the network architecture $\mathcal{F}_{\mathrm{NN}}(d_{\mathcal{Y}}, L, p, M)$ defined in equation 4 with parameters,*

$$M \geq \sqrt{d_{\mathcal{Y}}} L_{E_{\mathcal{Y}}^n} R_{\mathcal{Y}}, \ and \ Lp \geq \left\lceil d_{\mathcal{Y}}^{\frac{4-d_{\mathcal{X}}}{4+2d_{\mathcal{X}}}} n^{\frac{d_{\mathcal{X}}}{4+2d_{\mathcal{X}}}} \right\rceil. \tag{5}$$

*Then we have*

$$\mathcal{E}_{gen}(\Gamma_{\mathrm{NN}}) \lesssim L_{\Phi}^2 \log(L_{\Phi}) d_{\mathcal{Y}}^{\frac{8+d_{\mathcal{X}}}{2+d_{\mathcal{X}}}} n^{-\frac{2}{2+d_{\mathcal{X}}}} \log n + \mathcal{E}_{noise,proj}, \tag{6}$$

*where $\lesssim$ contains constants that depend on $d_{\mathcal{X}}, L_{E_{\mathcal{Y}}^n}, L_{E_{\mathcal{X}}^n}, L_{D_{\mathcal{Y}}^n}, L_{D_{\mathcal{X}}^n}$ and $R_{\mathcal{X}}$.*

*Remark* 3. The aforementioned results demonstrate that by selecting an appropriate width and depth for the DNN, the generalization error can be broken down into three components: the generalization error of learning the finite-dimensional operator $\Gamma$, the projection error of the encoders/decoders, and the noise. Comparing to previous results Liu et al. (2022) under the Hilbert space setting, our estimates show that the noise term in the generalization bound is non-decaying without the inner-product structure in the Banach space setting. This is mainly caused by circumventing the inner-product structure via triangle inequalities in the proof. As the number of samples $n$ increases, the generalization error decreases exponentially. Although the presence of $d_{\mathcal{X}}$ in the exponent of the sample complexity $n$ initially appears pessimistic, we will demonstrate that it can be eliminated when the input data $\Omega_{\mathcal{X}}$ of the target operator exhibits a low-dimensional data structure or when the target operator itself has a low-complexity structure. These assumptions are often satisfied for specific PDE operators with appropriate encoders. These results also imply that when $d_{\mathcal{X}}$ is large, the neural network width $p$ does not need to increase as the output dimension $d_{\mathcal{Y}}$ increases. The main difference between Theorem 1 and Theorem 2 lies in the different neural network architectures $\mathcal{F}_{\mathrm{NN}}(d_{\mathcal{Y}}, L, p, K, \kappa, M)$ and $\mathcal{F}_{\mathrm{NN}}(d_{\mathcal{Y}}, L, p, M)$. As a consequence, Theorem 2 has a smaller asymptotic lower bound $\Omega(n^{1/2})$ of the neural network width $p$ in the large $d_{\mathcal{X}}$ regime, whereas the asymptotic lower bound is $\Omega(n)$ in Theorem 1.

**Estimates with special data and operator structures**

The generalization error estimates presented in Theorems 1-2 are effective when the input dimension $d_{\mathcal{X}}$ is relatively small. However, in practical scenarios, it often requires numerous bases to reduce the encoder/decoder projection error, resulting in a large value for $d_{\mathcal{X}}$. Consequently, the decay rate of the generalization error as indicated in Theorems 1-2 becomes stagnant due to its exponential dependence on $d_{\mathcal{X}}$.

Nevertheless, it is often assumed that the high-dimensional data lie within the vicinity of a low-dimensional manifold by the famous "manifold hypothesis". Specifically, we assume that the encoded vectors $u$ lie on a $d_0$-dimensional manifold with $d_0 \ll d_{\mathcal{X}}$. Such a data distribution has been observed in many applications, including PDE solution set, manifold learning, and image recognition. This assumption is formulated as follows.

**Assumption 5.** Let $d_0 < d_{\mathcal{X}} \in \mathbb{N}$. Suppose there exists an encoder $E_{\mathcal{X}} : \mathcal{X} \to \mathbb{R}^{d_{\mathcal{X}}}$ such that $\{E_{\mathcal{X}}(u) \mid u \in \Omega_{\mathcal{X}}\}$ lies in a smooth $d_0$-dimensional Riemannian manifold $\mathcal{M}$ that is isometrically embedded in $\mathbb{R}^{d_{\mathcal{X}}}$. The *reach* Niyogi et al. (2008) of $\mathcal{M}$ is $\tau > 0$.

Under Assumption 5, the input data set exhibits a low intrinsic dimensionality. However, this may not hold for the output data set that is perturbed by noise. The reach of a manifold is the smallest osculating circle radius on the manifold. A manifold with large reach avoids rapid change and may be easier to learn by neural networks. In the following, we aim to demonstrate that the DNN naturally adjusts to the low-dimensional characteristics of the data set. As a result, the estimation error of the network depends solely on the intrinsic dimension $d_0$, rather than the larger ambient dimension $d_{\mathcal{X}}$. We present the following result to support this claim.

**Theorem 3.** *Suppose Assumptions 1-4, and Assumption 5 hold. Let $\Gamma_{\mathrm{NN}}$ be the minimizer of the optimization problem equation 1 with the network architecture $\mathcal{F}_{\mathrm{NN}}(d_{\mathcal{Y}}, L, p, M)$ defined in equation 4 with parameters*

$$L = \Omega(\tilde{L} \log \tilde{L}), p = \Omega(d_{\mathcal{X}} d_{\mathcal{Y}} \tilde{p} \log \tilde{p}), \quad M \geq \sqrt{d_{\mathcal{Y}}} L_{E_{\mathcal{Y}}^n} R_{\mathcal{Y}}, \tag{7}$$

where $\tilde{L}, \tilde{p} > 0$ are integers such that $\tilde{L}\tilde{p} \geq \left\lceil d_{\mathcal{Y}}^{\frac{-3d_0}{4+2d_0}} n^{\frac{d_0}{4+2d_0}} \right\rceil$. Then we have

$$\mathcal{E}_{gen}(\Gamma_{\mathrm{NN}}) \lesssim L_\Phi^2 \log(L_\Phi) d_{\mathcal{Y}}^{\frac{8+d_0}{2+d_0}} n^{-\frac{2}{2+d_0}} \log^6 n + \mathcal{E}_{noise,proj}, \tag{8}$$

where the constants in $\lesssim$ and $\Omega(\cdot)$ solely depend on $d_0, \log d_{\mathcal{X}}, R_{\mathcal{X}}, L_{E_{\mathcal{X}}^n}, L_{E_{\mathcal{Y}}^n}, L_{D_{\mathcal{X}}^n}, L_{D_{\mathcal{Y}}^n}, \tau$, the surface area of $\mathcal{M}$.

It is important to note that the estimate equation 8 depends at most polynomially on $d_{\mathcal{X}}$ and $d_{\mathcal{Y}}$. The rate of decay with respect to the sample size is no longer influenced by the ambient input dimension $d_{\mathcal{X}}$. Thus, our findings indicate that the CoD can be mitigated through the utilization of the "manifold hypothesis." To effectively capture the low-dimensional manifold structure of the data, the width of the DNN should be on the order of $O(d_{\mathcal{X}})$. Additionally, another characteristic often observed in PDE problems is the low complexity of the target operator. This holds true when the target operator is composed of several alternating sequences of a few linear and nonlinear transformations with only a small number of inputs. We quantify the notion of low-complexity operators in the following context.

**Assumption 6.** Let $0 < d_0 \leq d_{\mathcal{X}}$. Assume there exists $E_{\mathcal{X}}, D_{\mathcal{X}}, E_{\mathcal{Y}}, D_{\mathcal{Y}}$ such that for any $u \in \Omega_{\mathcal{X}}$, we have

$$\Pi_{\mathcal{Y},d_{\mathcal{Y}}} \circ \Phi(u) = D_{\mathcal{Y}} \circ g \circ E_{\mathcal{X}}(u),$$

where $g : \mathbb{R}^{d_{\mathcal{X}}} \to \mathbb{R}^{d_{\mathcal{Y}}}$ is defined as

$$g(a) = \left[ g_1(V_1^\top a), \cdots, g_{d_{\mathcal{Y}}}(V_{d_{\mathcal{Y}}}^\top a) \right],$$

where the matrix is $V_k \in \mathbb{R}^{d_{\mathcal{X}} \times d_0}$ and the real valued function is $g_k : \mathbb{R}^{d_0} \to \mathbb{R}$ for $k = 1, \ldots, d_{\mathcal{Y}}$. See an illustration in (44).

In Assumption 6, when $d_0 = 1$ and $g_1 = \cdots = g_{d_{\mathcal{Y}}}$, $g(a)$ is the composition of a pointwise nonlinear transform and a linear transform on $a$. In particular, Assumption 6 holds for any linear maps.

**Theorem 4.** *Suppose Assumptions 1-4, and Assumption 6 hold. Let $\Gamma_{\mathrm{NN}}$ be the minimizer of the optimization problem (1) with the network architecture $\mathcal{F}_{\mathrm{NN}}(d_{\mathcal{Y}}, L, p, M)$ defined in (4) with parameters*

$$Lp = \Omega\left( d_{\mathcal{Y}}^{\frac{4-d_0}{4+2d_0}} n^{\frac{d_0}{4+2d_0}} \right), M \geq \sqrt{d_{\mathcal{Y}}} L_{E_{\mathcal{Y}}^n} R_{\mathcal{Y}}.$$

*Then we have*

$$\mathcal{E}_{gen}(\Gamma_{\mathrm{NN}}) \lesssim L_\Phi^2 \log(L_\Phi) d_{\mathcal{Y}}^{\frac{8+d_0}{2+d_0}} n^{-\frac{2}{2+d_0}} \log n + \mathcal{E}_{noise,proj}, \tag{9}$$

where the constants in $\lesssim$ and $\Omega(\cdot)$ solely depend on $d_0, R_{\mathcal{X}}, R_{\mathcal{Y}}, L_{E_{\mathcal{X}}^n}, L_{E_{\mathcal{Y}}^n}, L_{D_{\mathcal{X}}^n}, L_{D_{\mathcal{Y}}^n}$.

*Remark* 4. Under Assumption 6, our result indicates that the CoD can be mitigated to a cost $\mathcal{O}(n^{\frac{-2}{2+d_0}})$ because the main task of DNNs is to learn the nonlinear transforms $g_1, \cdots, g_{d_{\mathcal{Y}}}$ that are functions over $\mathbb{R}^{d_0}$.

In practice, a PDE operator might be the repeated composition of operators in Assumption 6. This motivates a more general low-complexity assumption below.

**Assumption 7.** Let $0 < d_1, \ldots, d_k \leq d_{\mathcal{X}}$ and $0 < \ell_0, \ldots, \ell_k \leq \min\{d_{\mathcal{X}}, d_{\mathcal{Y}}\}$ with $\ell_0 = d_{\mathcal{X}}$ and $\ell_k = d_{\mathcal{Y}}$. Assume there exists $E_{\mathcal{X}}, D_{\mathcal{X}}, E_{\mathcal{Y}}, D_{\mathcal{Y}}$ such that for any $u \in \Omega_{\mathcal{X}}$, we have

$$\Pi_{\mathcal{Y},d_{\mathcal{Y}}} \circ \Phi(u) = D_{\mathcal{Y}} \circ G^k \circ \cdots \circ G^1 \circ E_{\mathcal{X}}(u),$$

where $G^i : \mathbb{R}^{\ell_{i-1}} \to \mathbb{R}^{\ell_i}$ is defined as

$$G^i(a) = \left[ g_1^i((V_1^i)^\top a), \cdots, g_{\ell_i}^i((V_{\ell_i}^i)^\top a) \right],$$

where the matrix is $V_j^i \in \mathbb{R}^{d_i \times \ell_{i-1}}$ and the real valued function is $g_j^i : \mathbb{R}^{d_i} \to \mathbb{R}$ for $j = 1, \ldots, \ell_i$, $i = 1, \ldots, k$. See an illustration in (45).

**Theorem 5.** *Suppose Assumptions 1-4, and Assumption 7 hold. Let $\Gamma_{\text{NN}}$ be the minimizer of the optimization (1) with the network architecture $\mathcal{F}_{\text{NN}}(d_{\mathcal{Y}}, kL, p, M)$ defined in equation 4 with parameters*

$$Lp = \Omega\left(d_{\mathcal{Y}}^{\frac{4-d_{max}}{4+2d_{max}}} n^{\frac{d_{max}}{4+2d_{max}}}\right), M \geq \sqrt{\ell_{max}} L_{E_{\mathcal{Y}}^n} R_{\mathcal{Y}},$$

*where $d_{max} = \max\{d_i\}_{i=1}^k$ and $\ell_{max} = \max\{\ell_i\}_{i=1}^k$. Then we have*

$$\mathcal{E}_{gen}(\Gamma_{\text{NN}}) \lesssim L_{\Phi}^2 \log(L_{\Phi}) \ell_{max}^{\frac{8+d_{max}}{2+d_{max}}} n^{-\frac{2}{2+d_{max}}} \log n + \mathcal{E}_{noise,proj},$$

*where the constants in $\lesssim$ and $\Omega(\cdot)$ solely depend on $k, d_{max}, \ell_{max}, R_{\mathcal{X}}, L_{E_{\mathcal{X}}^n}, L_{E_{\mathcal{Y}}^n}, L_{D_{\mathcal{X}}^n}, L_{D_{\mathcal{Y}}^n}$.*

**Discretization invariant neural networks**

In this subsection, we demonstrate that our main results also apply to neural networks with the discretization invariant property. A neural network is considered discretization invariant if it can be trained and evaluated on data that are discretized in various formats. For example, the input data $\mathbf{u}_i, i = 1, \ldots, n$ may consist of images with different resolutions, or $\mathbf{u}_i = [u_i(x_1), \ldots, u_i(x_{s_i})]$ representing the values of $u_i$ sampled at different locations. Neural networks inherently have fixed input and output sizes, making them incompatible for direct training on a data set $\{(\mathbf{u}_i, \mathbf{v}_i), i = 1, \ldots, n\}$ where the data pairs $(\mathbf{u}_i \in \mathbb{R}^{d_i}, \mathbf{v}_i \in \mathbb{R}^{d_i})$ have different resolutions $d_i, i = 1, \ldots, n$. Modifications of the encoders are required to map inputs of varying resolutions to a uniform Euclidean space. This can be achieved through linear interpolation or data-driven methods such as nonlinear integral transforms Ong et al. (2022).

Our previous analysis assumes that the data $(u_i, v_i) \in \mathcal{X} \times \mathcal{Y}$ is mapped to discretized data $(\mathbf{u}_i, \mathbf{v}_i) \in \mathbb{R}^{d_{\mathcal{X}}} \times \mathbb{R}^{d_{\mathcal{Y}}}$ using the encoders $E_{\mathcal{X}}^n$ and $E_{\mathcal{Y}}^n$. Now, let us consider the case where the new discretized data $(\mathbf{u}_i, \mathbf{v}_i) \in \mathbb{R}^{s_i} \times \mathbb{R}^{s_i}$ are vectors tabulating function values as follows:

$$\mathbf{u}_i = \begin{bmatrix} u_i(x_1^i) & u_i(x_2^i) & \ldots & u_i(x_{s_i}^i) \end{bmatrix}, \quad \mathbf{v}_i = \begin{bmatrix} v_i(x_1^i) & v_i(x_2^i) & \ldots & v_i(x_{s_i}^i) \end{bmatrix}. \tag{10}$$

The sampling locations $\mathbf{x}^i := [x_1^i, \ldots, x_{s_i}^i]$ are allowed to be different for each data pair $(\mathbf{u}_i, \mathbf{v}_i)$. We can now define the sampling operator on the location $\mathbf{x}^i$ as $P_{\mathbf{x}^i} : u \mapsto \mathbf{u}(\mathbf{x}^i)$, where $\mathbf{u}(\mathbf{x}^i) := \begin{bmatrix} u(x_1^i) & u(x_2^i) & \ldots & u(x_{s_i}^i) \end{bmatrix}$. For the sake of simplicity, we assume that the sampling locations are equally spaced grid points, denoted as $s_i = (r_i + 1)^d$, where $r_i + 1$ represents the number of grid points in each dimension. To achieve the discretization invariance, we consider the following interpolation operator $I_{\mathbf{x}^i} : \mathbf{u}(\mathbf{x}^i) \mapsto \tilde{u}$, where $\tilde{u}$ represents the multivariate Lagrangian polynomials (refer to Leaf & Kaper (1974) for more details). Subsequently, we map the Lagrangian polynomials to their discretization on a uniformly spaced grid mesh $\hat{\mathbf{x}} \in \mathbb{R}^{d_{\mathcal{X}}}$ using the sampling operator $P_{\hat{\mathbf{x}}}$. Here $d_{\mathcal{X}} = (r+1)^d$ and $r$ is the highest degree among the Lagrangian polynomials $\tilde{u}$. We further assume that the grid points $\mathbf{x}^i$ of all given discretized data are subsets of $\hat{\mathbf{x}}$. We can then construct a discretization-invariant encoder as follows:

$$E_{\mathcal{X}}^i = P_{\hat{\mathbf{x}}} \circ I_{\mathbf{x}^i} \circ P_{\mathbf{x}^i}.$$

We can define the encoder $E_{\mathcal{Y}}^i$ in a similar manner. The aforementioned discussion can be summarized in the following proposition:

**Proposition 1.** *Suppose that the discretized data $\{(\mathbf{u}_i, \mathbf{v}_i), i = 1, \ldots, n\}$ defined in equation 10 are images of a sampling operator $P_{\mathbf{x}^i}$ applied to smooth functions $(u_i, v_i)$, and the sampling locations $\mathbf{x}^i$ are equally spaced grid points with grid size $h$. Let $\hat{\mathbf{x}} \in \mathbb{R}^{d_{\mathcal{X}}}$ represent equally spaced grid points that are denser than all $\mathbf{x}^i, i = 1, \ldots, n$ with $d_{\mathcal{X}} = (r+1)^d$. Define the encoder $E_{\mathcal{X}}^i = E_{\mathcal{Y}}^i = P_{\hat{\mathbf{x}}} \circ I_{\mathbf{x}^i} \circ P_{\mathbf{x}^i}$, and decoder $D_{\mathcal{X}}^i = D_{\mathcal{Y}}^i = I_{\hat{\mathbf{x}}}$. Then the encoding error can be bounded as the following:*

$$\mathbb{E}_u\left[\|\Pi_{\mathcal{X}, d_{\mathcal{X}}}^i(u) - u\|_{\infty}^2\right] \leq Ch^{2r}\|u\|_{C^{r+1}}^2, \quad \mathbb{E}_v\left[\|\Pi_{\mathcal{Y}, d_{\mathcal{Y}}}^i(v) - v\|_{\infty}^2\right] \leq Ch^{2r}\|v\|_{C^{r+1}}^2, \tag{11}$$

*where $C > 0$ is an absolution constant, $\Pi_{\mathcal{X}, d_{\mathcal{X}}}^i := D_{\mathcal{X}}^i \circ E_{\mathcal{X}}^i$ and $\Pi_{\mathcal{Y}, d_{\mathcal{Y}}}^i := D_{\mathcal{Y}}^i \circ E_{\mathcal{Y}}^i$.*

*Proof.* This result follows directly from the principles of Multivariate Lagrangian interpolation and Theorem 3.2 in Leaf & Kaper (1974). $\square$

*Remark* 5. To simplify the analysis, we focus on the $L^\infty$ norm in equation 11. However, it is worth noting that $L^p$ estimates can be easily derived by utilizing $L^p$ space embedding techniques. Furthermore, $C^r$ estimates can be obtained through the proof of Theorem 3.2 in Leaf & Kaper (1974). By observing that the discretization invariant encoder and decoder in Proposition 1 satisfy Assumption 2 and Assumption 4, we can conclude that our main results are applicable to discretization invariant neural networks. In this section, we have solely considered polynomial interpolation encoders, which require the input data to possess a sufficient degree of smoothness and for all training data to be discretized on a finer mesh than the encoding space $\mathbb{R}^{d_{\mathcal{X}}}$. The analysis of more sophisticated nonlinear encoders and discretization invariant neural networks is a topic for future research.

In the subsequent sections, we will observe that numerous operators encountered in PDE problems can be expressed as compositions of low-complexity operators, as stated in Assumption 6 or Assumption 7. Consequently, deep operator learning provides means to alleviate the curse of dimensionality, as confirmed by Theorem 4 or its more general form, as presented in Theorem 5.

## 3 Explicit complexity bounds for various PDE operator learning

In practical scenarios, enforcing the uniform bound constraint in architecture (3) is often inconvenient. As a result, the preferred implementation choice is architecture (4). Therefore, in this section, we will solely focus on architecture (4). In this section, we will provide five examples of PDEs where the input space $\mathcal{X}$ and output space $\mathcal{Y}$ are not Hilbert. For simplicity, we assume that the computational domain for all PDEs is $\Omega = [-1, 1]^d$. Additionally, we assume that the input space $\mathcal{X}$ exhibits Hölder regularity. In other words, all inputs possess a bounded Hölder norm $\|\cdot\|_{C^s}$, where $s > 0$. The Hölder norm is defined as $\|f\|_{C^s} = \|f\|_{C^k} + \max_{\beta=k} |D^\beta f|_{C^{0,\alpha}}$, where $s = k + \alpha$, $k$ is an integer, $0 < \alpha < 1$ and $|\cdot|_{C^{0,\alpha}}$ represents the $\alpha$-Hölder semi-norm $|f|_{C^{0,\alpha}} = \sup_{x \neq y} \frac{|f(x)-f(y)|}{\|x-y\|^\alpha}$. It can be shown that the output space $\mathcal{Y}$ also admits Hölder regularity for all examples considered in this section. Similar results can be derived when both the input space and output space have Sobolev regularity. Consequently, we can employ the standard spectral method as the encoder/decoder for both the input and output spaces. Specifically, the encoder $E_{\mathcal{X}}^n$ maps $u \in \mathcal{X}$ to the space $P_d^r$, which represents the product of univariate polynomials with a degree less than $r$. As a result, the input dimension is thus $d_{\mathcal{X}} = \dim P_d^r = r^d$. We then assign $L^p$-norm ($p > 1$) to both the input space $\mathcal{X}$ and output space $\mathcal{Y}$. The encoder/decoder projection error for both $\mathcal{X}$ and $\mathcal{Y}$ can be derived using the following lemma from Schultz (1969).

**Lemma 1** (Theorem 4.3 (ii) of Schultz (1969)). *Let an integer $k \geq 0$ and $0 < \alpha < 1$. For any $f \in C^s([-1,1]^d)$ with $s = k + \alpha$, denote by $\tilde{f}$ its spectral approximation in $P_d^r$, there holds*

$$\|f - \tilde{f}\|_\infty \leq C_d \|f\|_{C^s} r^{-s}.$$

We can then bound the projection error

$$\|\Pi_{\mathcal{X}, d_{\mathcal{X}}}^n u - u\|_{L^p([-1,1]^d)}^p = \int_{[-1,1]^d} |u - \tilde{u}|^p dx \quad \leq C_d^p 2^d \|u\|_{C^s}^p r^{-ps} \leq C_d^p 2^d \|u\|_{C^s}^p d_{\mathcal{X}}^{-\frac{ps}{d}}.$$

Therefore,

$$\|\Pi_{\mathcal{X}, d_{\mathcal{X}}}^n u - u\|_{L^p([-1,1]^d)}^2 \leq C_d^2 2^{2d/p} \|u\|_{C^s}^2 d_{\mathcal{X}}^{-\frac{2s}{d}}. \tag{12}$$

Similarly, we can also derive that

$$\|\Pi_{\mathcal{Y}, d_{\mathcal{Y}}}^n (w) - w\|_{L^p([-1,1]^d)}^2 \leq C_d^2 2^{2d/p} \|u\|_{C^t}^2 d_{\mathcal{Y}}^{-\frac{2t}{d}} L_\Phi^2, \tag{13}$$

given that the output $w = \Phi(u)$ is in $C^t$ for some $t > 0$.

In the following, we present several examples of PDEs that satisfy different assumptions, including the low-dimensional Assumption 5, the low-complexity Assumption 6, and Assumption 7. In particular, the solution operators of Poisson equation, parabolic equation, and transport equation are linear operators, implying that Assumption 6 is satisfied with $g_i$'s being the identity functions with $d_0 = 1$. The solution operator of Burgers

equation is the composition of multiple numerical integration, the pointwise evaluation of an exponential function $g_j^1(\cdot) = \exp(\cdot)$, and the pointwise division $g_j^2(a, b) = a/b$. It thus satisfies Assumption 7 with $d_1 = 1$ and $d_2 = 2$. In parametric equations, we consider the forward operator that maps a media function $a(x)$ to the solution $u$. In most applications of such forward maps, the media function $a(x)$ represents natural images, such as CT scans for breast cancer diagnosis. Therefore, it is often assumed that Assumption 5 holds.

## 3.1 Poisson equation

Consider the Poisson equation which seeks $u$ such that

$$\Delta u = f, \tag{14}$$

where $x \in \mathbb{R}^d$, and $|u(x)| \to 0$ as $|x| \to \infty$. The fundamental solution of equation 14 is given as

$$\Psi(x) = \begin{cases} \frac{1}{2\pi} \ln |x|, & \text{for } d = 2, \\ \frac{-1}{w_d} |x|^{2-d}, & \text{for } d \geq 3, \end{cases}$$

where $w_d$ is the surface area of a unit ball in $\mathbb{R}^d$. Assume that the source $f(x)$ is a smooth function compactly supported in $\mathbb{R}^d$. There exists a unique solution to equation 14 given by $u(x) = \Psi * f$. Notice that the solution map $f \mapsto u$ is a convolution with the fundamental solution, $u(x) = \Psi * f$. To show the solution operator is Lipschitz, we assume the sources $f, g \in C^k(\mathbb{R}^d)$ with compact support and apply Young's inequality to get

$$\|u - v\|_{C^k(\mathbb{R}^d)} = \|D^k(u-v)\|_{L^\infty(\mathbb{R}^d)} = \|\Psi * D^k(f-g)\|_{L^\infty(\mathbb{R}^d)} \leq \|\Psi\|_{L^p(\mathbb{R}^d)} \|f - g\|_{C^k(\Omega)} |\Omega|^{1/q}, \tag{15}$$

where $p, q \geq 1$ so that $1/p + 1/q = 1$. Here $\Omega$ is the support of $f$ and $g$.

For the Poisson equation (14) on an unbounded domain, the computation is often implemented over a truncated finite domain $\Omega$. For simplicity, we assume the source condition $f$ is randomly generated in the space $C^k(\Omega)$ from a random measure $\gamma$. Since the solution $u$ is a convolution of source $f$ with a smooth kernel, both $f$ and $u$ are in $C^k(\Omega)$.

We then choose the encoder and decoder to be the spectral method. Applying equation 12, the encoder and decoder error of the input space can be calculated as follows

$$\mathbb{E}_f \left[ \|\Pi_{\mathcal{X}, d_{\mathcal{X}}}^n(f) - f\|_{L^p(\Omega)}^2 \right] \leq C_{d,p} d_{\mathcal{X}}^{-\frac{2k}{d}} \mathbb{E}_f \left[ \|f\|_{C^k(\Omega)}^2 \right].$$

Similarly, applying Lemma 1 and equation 15, the encoder and decoder error of the output space is

$$\mathbb{E}_{\mathcal{S}} \mathbb{E}_{f \sim \gamma} \left[ \|\Pi_{\mathcal{Y}, d_{\mathcal{Y}}}^n(u) - u\|_{L^p(\Omega)}^2 \right] \leq C_{d,p} d_{\mathcal{Y}}^{-\frac{2k}{d}} \mathbb{E}_f \left[ \|\Psi * f\|_{C^k(\Omega)}^2 \right] \leq C_{d,p,\Omega} d_{\mathcal{Y}}^{-\frac{2k}{d}} \mathbb{E}_f \left[ \|f\|_{C^k(\Omega)}^2 \right].$$

Notice that the solution $u(y) = \int_{\mathbb{R}^d} \Psi(y-x)f(x)dx$ is a linear integral transform of $f$, and that all linear maps are special cases of Assumption 6 with $g$ being the identity map. In particular, Assumption 6 thus holds true by setting the column vector $V_k$ as the numerical integration weight of $\Psi(x - y_k)$, and setting $g_k$'s as the identity map with $d_0 = 1$ for $k = 1, \cdots, d_{\mathcal{Y}}$. By applying Theorem 4, we obtain that

$$\mathbb{E}_{\mathcal{S}} \mathbb{E}_f \|D_{\mathcal{Y}}^n \circ \Gamma_{\text{NN}} \circ E_{\mathcal{X}}^n(f) - \Phi(f)\|_{L^p(\Omega)}^2 \lesssim r^{3d} n^{-2/3} \log n + (\sigma^2 + n^{-1}) + r^{-2k} \mathbb{E}_f \left[ \|f\|_{C^k(\Omega)}^2 \right], \tag{16}$$

where the input dimension $d_{\mathcal{X}} = d_{\mathcal{Y}} = r^d$ and $\lesssim$ contains constants that depend on $d_{\mathcal{X}}$, $d$, $p$ and $|\Omega|$.

*Remark* 6. The above result equation 16 suggests that the generalization error is small if we have a large number of samples, a small noise, and a good regularity of the input samples. Importantly, the decay rate with respect to the number of samples is independent from the encoding dimension $d_{\mathcal{X}}$ or $d_{\mathcal{Y}}$.

## 3.2 Parabolic equation

We consider the following parabolic equation that seeks $u(x, t)$ such that

$$\begin{cases} u_t - \Delta u = 0 & \text{in } \mathbb{R}^d \times (0, \infty), \\ u = g & \text{on } \mathbb{R}^d \times \{t = 0\}. \end{cases} \tag{17}$$

The fundamental solution to equation 17 is given by $\Lambda(x,t) = (4\pi t)^{-d/2} e^{-\frac{|x|^2}{4t}}$ for $x \in \mathbb{R}^d, t > 0$. The solution map $g(\cdot) \mapsto u(T,\cdot)$ can be expressed as a convolution with the fundamental solution $u(\cdot, T) = \Lambda(\cdot, T) * g$, where $T$ is the terminal time. Applying Young's inequality, the Lipschitz constant is $\|\Lambda(\cdot, T)\|_p$, where $1 \le p \le \infty$. As an example, we can explicitly calculate this number in 3D as $\|\Lambda(\cdot, T)\|_p = p^{\frac{3}{2p}}$. For the parabolic equation (17), we consider a truncated finite computation domain $\Omega \times [0, T]$ and assume an initial condition $g \in C^k(\Omega)$. Due to the similar convolution structure of the solution map compared to the Poisson equation, we can obtain a similar result by applying Theorem 4.

$$\mathbb{E}_{\mathcal{S}}\mathbb{E}_g \|D_{\mathcal{Y}}^n \circ \Gamma_{\text{NN}} \circ E_{\mathcal{X}}^n(g) - \Phi(g)\|_{L^p(\Omega)}^2 \lesssim r^{3d} n^{-2/3} \log n + (\sigma^2 + n^{-1}) + r^{-2k}\mathbb{E}_g\left[\|g\|_{C^k(\Omega)}^2\right], \tag{18}$$

where the encoding dimension $d_{\mathcal{X}} = d_{\mathcal{Y}} = r^d$, the symbol "$\lesssim$" denotes that the expression on the left-hand side is bounded by the expression on the right-hand side, where the constants involved depend on $d_{\mathcal{X}}$, $d$, $p$, and $|\Omega|$. The reduction of the CoD in the parabolic equation follows a similar approach as in the Poisson equation.

## 3.3 Transport equation

We consider the following transport equation that seeks $u$ such that

$$\begin{cases} u_t + a(x) \cdot \nabla u = 0 & \text{in } (0, \infty) \times \mathbb{R}^d, \\ u(0, x) = u_0(x) & \text{in } \mathbb{R}^d, \end{cases} \tag{19}$$

where $a(x)$ is the drift force field and $u_0(x)$ is the initial data. For convenience, we assume that the drift force field satisfies $a \in C^2(\mathbb{R}^d) \cap W^{1,\infty}(\mathbb{R}^d)$. By employing the classical theory of ordinary differential equations (ODE), we consider the initial value problem $\frac{dx(t)}{dt} = a(x(t))$, $x(0) = x$, which admits a unique solution for any $x \in \mathbb{R}^d$, $t \to x(t) = \varphi_t(x) \in C^1(\mathbb{R}; \mathbb{R}^d)$. Applying the Characteristic method, the solution of equation 19 is given by $u(t, x) := u_0(\varphi_t^{-1}(x))$. If we further assume that $u_0$ is randomly sampled with bounded $H^s$ norm, $s > \frac{3d}{2}$, then by Theorem 5 of Section 7.3 of Evans (2010), we have $u \in C^1([0, \infty); \mathbb{R}^d)$. More specifically, we have

$$\|u(T, \cdot)\|_{C^1(\mathbb{R}^d)} \le \|u_0\|_{H^s(\mathbb{R}^d)} C_{a,T,\Omega},$$

where $C_{a,T,\Omega} > 0$ is a constant that depends on the media $a$, terminal time $T$, and the support $\Omega$ of the initial data. Since the initial data has $C^1$ regularity, by equation 12 the encoder/decoder projection error of the input space is controlled via

$$\mathbb{E}_{u_0}\left[\|\Pi_{\mathcal{X}, d_{\mathcal{X}}}^n(u_0) - u_0\|_{L^p(\Omega)}^2\right] \le C_{d,p,\Omega} d_{\mathcal{X}}^{-\frac{2}{d}} \mathbb{E}_f\left[\|u_0\|_{C^1(\Omega)}^2\right].$$

Similarly, for the projection error of the output space, we have

$$\mathbb{E}_{\mathcal{S}}\mathbb{E}_{u \sim \Phi_\# \gamma}\left[\|\Pi_{\mathcal{Y}, d_{\mathcal{Y}}}^n(u) - u\|_{L^p(\Omega)}^2\right] \le C_{d,p,\Omega} d_{\mathcal{Y}}^{-\frac{2}{d}} \mathbb{E}_{u_0}\left[\|u(T)\|_{C^1(\Omega)}^2\right] \le C_{d,p,a,T,\Omega} d_{\mathcal{Y}}^{-\frac{2}{d}} \mathbb{E}_{u_0}\left[\|u_0\|_{H^s(\Omega)}^2\right].$$

We again use the spectral encoder/decoder so $d_{\mathcal{X}} = d_{\mathcal{Y}} = r^d$. Notice that solution $u(T, x) = u_0(\varphi_T^{-1}(x))$ is a translation of the initial data $u_0$ by $\varphi_T^{-1}$, which is a linear transform. Let $V \in \mathbb{R}^{d_{\mathcal{X}} \times d_{\mathcal{Y}}}$ be the corresponding permutation matrix that characterizes the translation by $\varphi_T^{-1}$, then $V_k^\top$ is the $k$-th row of $V$. Then by setting $g_k$'s as the identity map, Assumption 6 holds with $d_0 = 1$. Apply Theorem 4 to derive that

$$\mathbb{E}_{\mathcal{S}}\mathbb{E}_u \|D_{\mathcal{Y}}^n \circ \Gamma_{\text{NN}} \circ E_{\mathcal{X}}^n(u) - \Phi(u)\|_{L^p(\Omega)}^2 \lesssim r^{3d} n^{-2/3} \log n + (\sigma^2 + n^{-1}) + r^{-2}\mathbb{E}_g\left[\|u_0\|_{C^1(\Omega)}^2 + \|u_0\|_{H^s(\Omega)}^2\right], \tag{20}$$

where $\lesssim$ contains constants that depend on $d, p, a, r, T$ and $\Omega$. The CoD in transport equation is lessened according to equation 20 in the same manner as in the Poisson and parabolic equations.

### 3.4 Burgers equation

We consider the 1D Burgers equation with periodic boundary conditions:

$$\begin{cases} u_t + uu_x = \kappa u_{xx}, & \text{in } \mathbb{R} \times (0, \infty), \\ u(x, 0) = u_0(x), \\ u(-\pi, t) = u(\pi, t), \end{cases} \tag{21}$$

where $\kappa > 0$ is the viscosity constant. and we consider the solution map $u_0(\cdot) \mapsto u(T, \cdot)$. This solution map can be explicitly written using the Cole-Hopf transformation $u = \frac{-2\kappa v_x}{v}$ where the function $v$ is the solution to the following diffusion equation

$$\begin{cases} v_t = \kappa v_{xx} \\ v(x, 0) = v_0(x) = \exp\left(-\frac{1}{2\kappa} \int_{-\pi}^{x} u_0(s) ds\right). \end{cases}$$

The solution to the above diffusion equation is given by

$$v(x, T) = -2\kappa \frac{\int_{\mathbb{R}} \partial_x \mathcal{K}(x, y, T) v_0(y) dy}{\int_{\mathbb{R}} \mathcal{K}(x, y, T) v_0(y) dy}, \tag{22}$$

where the integration kernel $\mathcal{K}$ is defined as $\mathcal{K}(x, y, t) = \frac{1}{\sqrt{4\pi\kappa t}} \exp\left(\frac{-(x-y)^2}{4\pi t}\right)$. Although there will be no shock formed in the solution of viscous Burger equation, the solution may form a large gradient in finite time for certain initial data, which makes it extremely hard to be approximated by a NN. We assume that the terminal time $T$ is small enough so a large gradient is not formed yet. In fact, it is shown in Heywood & Xie (1997) (Theorem 1) that if $T \leq C\|u_0\|_{H^1}^{-4}$, then $\|u(\cdot, T)\|_{H^1} \leq C\|u_0\|_{H^1}$. We then assume an initial data $u_0$ is randomly sampled with a uniform bounded $H^1$ norm. By Sobolev embedding, we have

$$\|u_0\|_{C^{0,1/2}} \leq C\|u_0\|_{H^1}, \quad \|u(\cdot, T)\|_{C^{0,1/2}} \leq C\|u_0\|_{H^1},$$

By 12, we can control the encoder/decoder projection error for the initial data

$$\mathbb{E}_{u_0}\left[\|\Pi_{\mathcal{X}, d_{\mathcal{X}}}^n(u_0) - u_0\|_{L^p(\Omega)}^2\right] \leq C_{d,p} d_{\mathcal{X}}^{-1} \mathbb{E}_{u_0}\left[\|u_0\|_{H^1}^2\right].$$

Since the terminal solution $u(\cdot, T)$ has same regularity as the initial solution, by 13 we also have

$$\mathbb{E}_{u_0}\left[\|\Pi_{\mathcal{X}, d_{\mathcal{X}}}^n(u(\cdot, T)) - u(\cdot, T)\|_{L^p(\Omega)}^2\right] \leq C_{d,p} d_{\mathcal{Y}}^{-1} \mathbb{E}_{u_0}\left[\|u(\cdot, T)\|_{H^1}^2\right] \leq C_{d,p} d_{\mathcal{Y}}^{-1} \mathbb{E}_{u_0}\left[\|u_0\|_{H^1}^2\right].$$

Similarly, we can choose $d_{\mathcal{X}} = d_{\mathcal{Y}} = r$. The solution map is a composition of three mappings $u_0 \mapsto v_0$, $v_0 \mapsto v(\cdot, T)$ and $v(\cdot, T) \mapsto u(\cdot, T)$. More specifically, $v_0(x) = \exp\left(-\frac{1}{2\kappa} \int_{-\pi}^{x} u_0(s) ds\right)$ so we can set $V_k^1 = \mathbb{R}^{d_{\mathcal{X}} \times 1}$ as the numerical integration vector on $[-\pi, x_k]$ and $g_k^1(x) = \exp(\frac{-x}{2\kappa})$ for all $k = 1, \ldots, d_{\mathcal{Y}}$. For the second mapping $v_0 \mapsto v(\cdot, T)$ (c.f. 22), we set $V_k^2 \in \mathbb{R}^{d_{\mathcal{X}} \times 2}$ where the first row is the numerical integration with kernel $\partial_x \mathcal{K}$ and the second row is the numerical integration with kernel $\mathcal{K}$, and we let $g_k^2(x, y) = \frac{-2\kappa x}{y}$ for all $k = 1, \cdots, d_{\mathcal{X}}$. For the third mapping $u = \frac{-2\kappa v_x}{v}$, we can set $V_k^3 \in \mathbb{R}^{d_{\mathcal{X}} \times 2}$, where the first row is the $k$-row of the numerical differentiation matrix, and the second row is the Dirac-delta vector at $x_k$, and we let $g_k^3(x, y) = \frac{-2\kappa x}{y}$ for all $k = 1, \cdots, d_{\mathcal{Y}}$. Therefore, Assumption 7 holds with $d_{\max} = 2$ and $l_{\max} = d_{\mathcal{X}} = d_{\mathcal{Y}} = r$. Then apply Theorem 5 to derive that

$$\mathbb{E}_{\mathcal{S}} \mathbb{E}_u \|D_{\mathcal{Y}}^n \circ \Gamma_{\text{NN}} \circ E_{\mathcal{X}}^n(u_0) - u(\cdot, T)\|_{L^p(\Omega)}^2 \lesssim r^{5/2} n^{-1/2} \log n + (\sigma^2 + n^{-1}) + r^{-1} \mathbb{E}_g\left[\|u_0\|_{H^s(\Omega)}^2\right], \tag{23}$$

where $\lesssim$ contains constants that depend on $p, r$ and $T$. The CoD in Burgers equations is lessened according to equation 23 as well as in all other PDE examples.

### 3.5 Parametric elliptic equation

We consider the 2D elliptic equation with heterogeneous media in this subsection.

$$
\begin{cases}
-\mathrm{div}(a(x)\nabla_x u(x)) = 0\,, & \text{in } \Omega \subset \mathbb{R}^2, \\
u = f\,, & \text{on } \partial\Omega.
\end{cases}
\tag{24}
$$

The media coefficient $a(x)$ satisfies that $\alpha \leq a(x) \leq \beta$ for all $x \in \Omega$, where $\alpha$ and $\beta$ are positive constants. We further assume that $a(x) \in C^1(\Omega)$. We are interested NN approximation of the forward map $\Phi : a \mapsto u$ with a fixed boundary condition $f$, which has wide applications in inverse problems. The forward map is Lipschitz, see Appendix A.2. We apply Sobolev embedding and derive that $u \in C^{0,1/2}(\Omega)$. Since the parameter $a$ has $C^1$ regularity, the encoder/decoder projection error of the input space is controlled

$$
\mathbb{E}_a \left[ \|\Pi^n_{\mathcal{X},d_{\mathcal{X}}}(a) - a\|^2_{L^p(\Omega)} \right] \leq C_p d_{\mathcal{X}}^{-1} \mathbb{E}_f \left[ \|a\|^2_{C^1(\Omega)} \right].
$$

The solution has $\frac{1}{2}$ Hölder regularity, so we have

$$
\mathbb{E}_{\mathcal{S}} \mathbb{E}_{u \sim \Phi_{\#}\gamma} \left[ \|\Pi^n_{\mathcal{Y},d_{\mathcal{Y}}}(u) - u\|^2_{L^p(\Omega)} \right] = \mathbb{E}_a \left[ \|\Pi^n_{\mathcal{Y},d_{\mathcal{Y}}}(u) - u\|_{L^p(\Omega)} \right]^2 \leq C_p d_{\mathcal{Y}}^{-\frac{1}{2}} \mathbb{E}_a \left[ \|u\|^2_{C^{0,1/2}(\Omega)} \right] \leq C_{p,\alpha,\beta,f} d_{\mathcal{Y}}^{-\frac{1}{2}}.
$$

We use the spectral encoder/decoder and choose $d_{\mathcal{X}} = d_{\mathcal{Y}} = r^2$. We further assume that the media functions $a(x)$ are randomly sampled on a smooth $d_0$-dimensional manifold. Applying Theorem 3, the generalization error is thus bounded by

$$
\mathbb{E}_{\mathcal{S}} \mathbb{E}_u \|D^n_{\mathcal{Y}} \circ \Gamma_{\mathrm{NN}} \circ E^n_{\mathcal{X}}(u) - \Phi(u)\|^2_{L^p(\Omega)} \lesssim d_{\mathcal{Y}}^{\frac{8+d_0}{2+d_0}} n^{-\frac{2}{2+d_0}} \log^6 n + (\sigma^2 + n^{-1}) + r^{-2} \mathbb{E}_a \left[ \|a\|^2_{C^1(\Omega)} \right] + r^{-1},
$$

where $\lesssim$ contains constants that depend on $p, \Omega, \alpha, \beta$ and $f$. Here $d_0$ is a constant that characterized the manifold dimension of the data set of media function $a(x)$. For instance, the 2D Shepp-Logan phantom Gach et al. (2008) contains multiple ellipsoids with different intensities thus the images in this data set lies on a manifold with a small $d_0$. The decay rate in terms of the number of samples $n$ solely depends on $d_0$, therefore the CoD of the parametric elliptic equations is mitigated.

## 4 Limitations and discussions

Our work focuses on exploring the efficacy of fully connected DNNs as surrogate models for solving general PDE problems. We provide an explicit estimation of the training sample complexity for generalization error. Notably, when the PDE solution lies in a low-dimensional manifold or the solution space exhibits low complexity, our estimate demonstrates a logarithmic dependence on the problem resolution, thereby reducing the CoD. Our findings offer a theoretical explanation for the improved performance of deep operator learning in PDE applications.

However, our work relies on the assumption of Lipschitz continuity for the target PDE operator. Consequently, our estimates may not be satisfactory if the Lipschitz constant is large. This limitation hampers the application of our theory to operator learning in PDE inverse problems, which focus on the solution-to-parameter map. Although the solution-to-parameter map is Lipschitz in many applications (e.g., electric impedance tomography, optical tomography, and inverse scattering), certain scenarios may feature an exponentially large Lipschitz constant, rendering our estimates less practical. Therefore, our results cannot fully explain the empirical success of PDE operator learning in such cases.

While our primary focus is on neural network approximation, determining suitable encoders and decoders with small encoding dimensions ($d_{\mathcal{X}}$ and $d_{\mathcal{Y}}$) remains a challenging task that we did not emphasize in this work. In Section 2.2, we analyze the naive interpolation as a discretization invariant encoder using a fully connected neural network architecture. However, this analysis is limited to cases where the training data is sampled on an equally spaced mesh and may not be applicable to more complex neural network architectures or situations where the data is not uniformly sampled. Investigating the discretization invariant properties of other neural networks, such as IAE-net Ong et al. (2022), FNO Li et al. (2021), and DeepONet, would be an interesting avenue for future research.

## Acknowledgements

K. C. and H. Y. were partially supported by the US National Science Foundation under awards DMS-2244988, DMS-2206333, and the Office of Naval Research Award N00014-23-1-2007. C. W. was partially supported by the National Science Foundation under awards DMS-2136380 and DMS-2206332.

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

## A    Appendix

### A.1    Proofs of the main theorems

*Proof of Theorem 1.* The $L^2$ squared error can be decomposed as

$$\mathbb{E}_{\mathcal{S}}\mathbb{E}_{u\sim\gamma}\left[\|D_{\mathcal{Y}}^n\circ\Gamma_{\mathrm{NN}}\circ E_{\mathcal{X}}^n(u)-\Phi(u)\|_{\mathcal{Y}}^2\right]$$
$$\leq 2\mathbb{E}_{\mathcal{S}}\mathbb{E}_{u\sim\gamma}\left[\|D_{\mathcal{Y}}^n\circ\Gamma_{\mathrm{NN}}\circ E_{\mathcal{X}}^n(u)-D_{\mathcal{Y}}^n\circ E_{\mathcal{Y}}^n\circ\Phi(u)\|_{\mathcal{Y}}^2\right]+2\mathbb{E}_{\mathcal{S}}\mathbb{E}_{u\sim\gamma}\left[\|D_{\mathcal{Y}}^n\circ E_{\mathcal{Y}}^n\circ\Phi(u)-\Phi(u)\|_{\mathcal{Y}}^2\right],$$

where the first term $\mathrm{I}=2\mathbb{E}_{\mathcal{S}}\mathbb{E}_{u\sim\gamma}\left[\|D_{\mathcal{Y}}^n\circ\Gamma_{\mathrm{NN}}\circ E_{\mathcal{X}}^n(u)-D_{\mathcal{Y}}^n\circ E_{\mathcal{Y}}^n\circ\Phi(u)\|_{\mathcal{Y}}^2\right]$ is the network estimation error in the $\mathcal{Y}$ space, and the second term $\mathrm{II}=2\mathbb{E}_{\mathcal{S}}\mathbb{E}_{u\sim\gamma}\left[\|D_{\mathcal{Y}}^n\circ E_{\mathcal{Y}}^n\circ\Phi(u)-\Phi(u)\|_{\mathcal{Y}}^2\right]$ is the empirical projection error, which can be rewritten as

$$\mathrm{II}=2\mathbb{E}_{\mathcal{S}}\mathbb{E}_{w\sim\Phi_{\#}\gamma}\left[\|\Pi_{\mathcal{Y},d_{\mathcal{Y}}}^n(w)-w\|_{\mathcal{Y}}^2\right]. \tag{25}$$

We aim to derive an upper bound of the first term I. First, note that the decoder $D_{\mathcal{Y}}^n$ is Lipschitz (Assumption 3). We have

$$\mathrm{I}=2\mathbb{E}_{\mathcal{S}}\mathbb{E}_{u\sim\gamma}\left[\|D_{\mathcal{Y}}^n\circ\Gamma_{\mathrm{NN}}\circ E_{\mathcal{X}}^n(u)-D_{\mathcal{Y}}^n\circ E_{\mathcal{Y}}^n\circ\Phi(u)\|_{\mathcal{Y}}^2\right]$$
$$\leq 2L_{D_{\mathcal{Y}}^n}^2\mathbb{E}_{\mathcal{S}}\mathbb{E}_{u\sim\gamma}\left[\|\Gamma_{\mathrm{NN}}\circ E_{\mathcal{X}}^n(u)-E_{\mathcal{Y}}^n\circ\Phi(u)\|_2^2\right].$$

Conditioned on the data set $\mathcal{S}_1$, we can obtain

$$\mathbb{E}_{\mathcal{S}_2}\mathbb{E}_{u\sim\gamma}\left[\|\Gamma_{\mathrm{NN}}\circ E_{\mathcal{X}}^n(u)-E_{\mathcal{Y}}^n\circ\Phi(u)\|_2^2\right]$$
$$=2\mathbb{E}_{\mathcal{S}_2}\left[\frac{1}{n}\sum_{i=n+1}^{2n}\|\Gamma_{\mathrm{NN}}\circ E_{\mathcal{X}}^n(u_i)-E_{\mathcal{Y}}^n\circ\Phi(u_i)\|_2^2\right]$$
$$+\mathbb{E}_{\mathcal{S}_2}\mathbb{E}_{u\sim\gamma}\left[\|\Gamma_{\mathrm{NN}}\circ E_{\mathcal{X}}^n(u)-E_{\mathcal{Y}}^n\circ\Phi(u)\|_2^2\right]-2\mathbb{E}_{\mathcal{S}_2}\left[\frac{1}{n}\sum_{i=n+1}^{2n}\|\Gamma_{\mathrm{NN}}\circ E_{\mathcal{X}}^n(u_i)-E_{\mathcal{Y}}^n\circ\Phi(u_i)\|_2^2\right] \tag{26}$$
$$=T_1+T_2,$$

where the first term $T_1 = 2\mathbb{E}_{\mathcal{S}_2}\left[\frac{1}{n}\sum_{i=n+1}^{2n}\|\Gamma_{\mathrm{NN}}\circ E_{\mathcal{X}}^n(u_i)-E_{\mathcal{Y}}^n\circ\Phi(u_i)\|_2^2\right]$ includes the DNN approximation error and the projection error in the $\mathcal{X}$ space, and the second term $T_2 = \mathbb{E}_{\mathcal{S}_2}\mathbb{E}_{u\sim\gamma}\left[\|\Gamma_{\mathrm{NN}}\circ E_{\mathcal{X}}^n(u)-E_{\mathcal{Y}}^n\circ\Phi(u)\|_2^2\right]-T_1$ captures the variance.

To obtain an upper bound of $T_1$, we apply triangle inequality to separate the noise from $T_1$

$$T_1\leq 2\mathbb{E}_{\mathcal{S}_2}\left[\frac{1}{n}\sum_{i=n+1}^{2n}\|\Gamma_{\mathrm{NN}}\circ E_{\mathcal{X}}^n(u_i)-E_{\mathcal{Y}}^n(v_i)\|_2^2\right]+2\mathbb{E}_{\mathcal{S}_2}\left[\frac{1}{n}\sum_{i=n+1}^{2n}\|E_{\mathcal{Y}}^n\circ\Phi(u_i)-E_{\mathcal{Y}}^n(v_i)\|_2^2\right].$$

Using the definition of $\Gamma_{\mathrm{NN}}$, we have

$$T_1\leq 2\mathbb{E}_{\mathcal{S}_2}\left[\inf_{\Gamma\in\mathrm{NN}}\frac{1}{n}\sum_{i=n+1}^{2n}\|\Gamma\circ E_{\mathcal{X}}^n(u_i)-E_{\mathcal{Y}}^n(v_i)\|_2^2\right]+2L_{E_{\mathcal{Y}}^n}^2\mathbb{E}_{\mathcal{S}_2}\frac{1}{n}\sum_{i=n+1}^{2n}\|\varepsilon_i\|_{\mathcal{Y}}^2.$$

Using Fatou's lemma, we have

$$T_1\leq 4\mathbb{E}_{\mathcal{S}_2}\left[\inf_{\Gamma\in\mathcal{F}_{\mathrm{NN}}}\frac{1}{n}\sum_{i=n+1}^{2n}\|\Gamma\circ E_{\mathcal{X}}^n(u_i)-E_{\mathcal{Y}}^n\circ\Phi(u_i)\|_2^2\right]+6L_{E_{\mathcal{Y}}^n}^2\mathbb{E}_{\mathcal{S}_2}\frac{1}{n}\sum_{i=n+1}^{2n}\|\varepsilon_i\|_{\mathcal{Y}}^2$$
$$\leq 4\inf_{\Gamma\in\mathcal{F}_{\mathrm{NN}}}\mathbb{E}_{\mathcal{S}_2}\left[\frac{1}{n}\sum_{i=n+1}^{2n}\|\Gamma\circ E_{\mathcal{X}}^n(u_i)-E_{\mathcal{Y}}^n\circ\Phi(u_i)\|_2^2\right]+6L_{E_{\mathcal{Y}}^n}^2\mathbb{E}_{\mathcal{S}_2}\frac{1}{n}\sum_{i=n+1}^{2n}\|\varepsilon_i\|_{\mathcal{Y}}^2 \tag{27}$$
$$=4\inf_{\Gamma\in\mathcal{F}_{\mathrm{NN}}}\mathbb{E}_u\left[\|\Gamma\circ E_{\mathcal{X}}^n(u)-E_{\mathcal{Y}}^n\circ\Phi(u)\|_2^2\right]+6L_{E_{\mathcal{Y}}^n}^2\mathbb{E}_{\mathcal{S}_2}\frac{1}{n}\sum_{i=n+1}^{2n}\|\varepsilon_i\|_{\mathcal{Y}}^2.$$

To bound the first term on the last line of Equation equation 27, we consider the discrete transform $\Gamma_d^n :=  E_{\mathcal{Y}}^n \circ \Phi \circ D_{\mathcal{X}}^n \in \mathcal{F}_{NN}$. Note that it is a vector filed that maps $\mathbb{R}^{d_{\mathcal{X}}}$ to $\mathbb{R}^{d_{\mathcal{Y}}}$, and by Assumption 1, 2, and 3 each component $h(\cdot)$ is a function supported on $[-B, B]^{d_{\mathcal{X}}}$ with Lipschitz constant $M := L_{D_{\mathcal{X}}^n} L_\Phi L_{E_{\mathcal{Y}}^n}$, where $B = R_{\mathcal{X}} L_{E_{\mathcal{X}}^n}$. This implies that each component $h$ has an infinity bound $\|h\|_\infty \leq R := BM = L_{D_{\mathcal{X}}^n} L_\Phi L_{E_{\mathcal{Y}}^n} R_{\mathcal{X}} L_{E_{\mathcal{X}}^n}$.

We now apply the following lemma to the component functions of $\Gamma_d^n$.

**Lemma 2.** *For any function $f \in W^{n,\infty}([-1,1]^d)$, and $\epsilon \in (0,1)$, we assume that $\|f\|_{W^{n,\infty}} \leq 1$. There exists a function $\tilde{f} \in \mathcal{F}_{NN}(1, L, p, K, \kappa, M)$ such that*

$$\|\tilde{f} - f\|_\infty < \epsilon,$$

*where the parameters of $\mathcal{F}_{NN}$ are chosen as*

$$
\begin{aligned}
L &= \Omega((n+d)\ln \epsilon^{-1} + n^2 \ln d + d^2), \quad p = \Omega(d^{d+n} \epsilon^{-\frac{d}{n}} n^{-d} 2^{d^2/n}), \\
K &= \Omega(n^{2-d} d^{d+n+2} 2^{\frac{d^2}{n}} \epsilon^{-\frac{d}{n}} \ln \epsilon), \quad \kappa = \Omega(M^2), \quad M = \Omega(d+n).
\end{aligned}
$$
(28)

*Here all constants hidden in $\Omega(\cdot)$ do not dependent on any parameters.*

*Proof.* This is a direct consequence of proof of Theorem 1 in Yarotsky (2017) for $F_{n,d}$. $\qquad\square$

Let $h_i : \mathbb{R}^{d_{\mathcal{X}}} \to \mathbb{R}, i = 1, \ldots, d_{\mathcal{Y}}$ be the components of $\Gamma_d^n$, then apply Lemma 2 to the rescaled component $\frac{1}{R} h_i(B \cdot)$ with $n = 1$. It can be derived that there exists $\tilde{h}_i \in \mathcal{F}_{NN}(1, L, \tilde{p}, K, \kappa, M)$ such that

$$\max_{x \in [-1,1]^{d_{\mathcal{X}}}} |\frac{1}{R} h_i(Bx) - \tilde{h}_i(x)| \leq \tilde{\varepsilon}_1,$$

with parameters chosen as in equation 28, with $n = 1$, $d = d_{\mathcal{X}}$, and $\epsilon = \tilde{\varepsilon}$. Using a change of variable, we obtain that

$$\max_{x \in [-B,B]^{d_{\mathcal{X}}}} |h_i(x) - R\tilde{h}_i(\frac{x}{B})| \leq R\tilde{\varepsilon}_1.$$

Assembling the neural networks $R\tilde{h}_i(\frac{\cdot}{B})$ together, we obtain an neural network $\tilde{\Gamma}_d^n \in \mathcal{F}_{NN}(d_{\mathcal{Y}}, L, p, K, \kappa, M)$ with $p = d_{\mathcal{Y}}\tilde{p}$, such that

$$\|\tilde{\Gamma}_d^n - \Gamma_d^n\|_\infty \leq \varepsilon_1,$$
(29)

Here the parameters of $\mathcal{F}_{NN}(d_{\mathcal{Y}}, L, p, K, \kappa, M)$ are chosen as

$$
\begin{aligned}
L &= \Omega(d_{\mathcal{X}} \ln \varepsilon_1^{-1}), \quad p = \Omega(d_{\mathcal{Y}} \varepsilon_1^{-d_{\mathcal{X}}} L_\Phi^{-d_{\mathcal{X}}} 2^{d_{\mathcal{X}}^2}), \\
K &= \Omega(pL), \quad \kappa = \Omega(M^2), \quad M \geq \sqrt{d_{\mathcal{Y}}} L_{E_{\mathcal{X}}^n} R_{\mathcal{Y}}.
\end{aligned}
$$
(30)

Here the constants in $\Omega$ may depend on $L_{D_{\mathcal{X}}^n}, L_{E_{\mathcal{X}}^n}, L_{E_{\mathcal{Y}}^n}$ and $R_{\mathcal{X}}$. Then we can develop an estimate of $T_1$ as follows.

$$
\begin{aligned}
&\inf_{\Gamma \in \mathcal{F}_{NN}} \mathbb{E}_u \left[ \|\Gamma \circ E_{\mathcal{X}}^n(u) - E_{\mathcal{Y}}^n \circ \Phi(u)\|_2^2 \right] \\
\leq& \mathbb{E}_u \left[ \|\tilde{\Gamma}_d^n \circ E_{\mathcal{X}}^n(u) - E_{\mathcal{Y}}^n \circ \Phi(u)\|_2^2 \right] \\
\leq& 2\mathbb{E}_u \left[ \|\tilde{\Gamma}_d^n \circ E_{\mathcal{X}}^n(u) - \Gamma_d^n \circ E_{\mathcal{X}}^n(u)\|_2^2 \right] + 2\mathbb{E}_u \left[ \|\Gamma_d^n \circ E_{\mathcal{X}}^n(u) - E_{\mathcal{Y}}^n \circ \Phi(u)\|_2^2 \right] \\
\leq& 2d_{\mathcal{Y}} \varepsilon_1^2 + 2\mathbb{E}_u \left[ \|\Gamma_d^n \circ E_{\mathcal{X}}^n(u) - E_{\mathcal{Y}}^n \circ \Phi(u)\|_2^2 \right],
\end{aligned}
$$
(31)

where we used the definition of infinimum in the first inequality, the triangle inequality in the second inequality, and the approximation equation 29 in the third inequality. Using the definition of $\Phi$, we obtain

$$
\begin{aligned}
&\inf_{\Gamma \in \mathcal{F}_{NN}} \mathbb{E}_u \left[ \|\Gamma \circ E_{\mathcal{X}}^n(u) - E_{\mathcal{Y}}^n \circ \Phi(u)\|_2^2 \right] \\
=& 2d_{\mathcal{Y}} \varepsilon_1^2 + 2\mathbb{E}_u \left[ \|E_{\mathcal{Y}}^n \circ \Phi \circ D_{\mathcal{X}}^n \circ E_{\mathcal{X}}^n(u) - E_{\mathcal{Y}}^n \circ \Phi(u)\|_2^2 \right] \\
\leq& 2d_{\mathcal{Y}} \varepsilon_1^2 + 2L_{E_{\mathcal{Y}}^n}^2 L_\Phi^2 \mathbb{E}_u \left[ \|D_{\mathcal{X}}^n \circ E_{\mathcal{X}}^n(u) - u\|_{\mathcal{X}}^2 \right] \\
=& 2d_{\mathcal{Y}} \varepsilon_1^2 + 2L_{E_{\mathcal{Y}}^n}^2 L_\Phi^2 \mathbb{E}_u \left[ \|\Pi_{\mathcal{X},d_{\mathcal{X}}}^n(u) - u\|_{\mathcal{X}}^2 \right],
\end{aligned}
$$
(32)

where we used the Lipschitz continuity of $\Phi$ and $E_{\mathcal{Y}}^n$ in the inequality above. Combining equation 32 and equation 27, and apply Assumption 4, we have

$$T_1 \leq 8d_{\mathcal{Y}}\varepsilon_1^2 + 8L_{E_{\mathcal{Y}}^n}^2 L_{\Phi}^2 \mathbb{E}_u \left[\|\Pi_{\mathcal{X},d_{\mathcal{X}}}^n(u) - u\|_{\mathcal{X}}^2\right] + 6L_{E_{\mathcal{Y}}^n}^2 \sigma^2. \tag{33}$$

To deal with the term $T_2$, we shall use the covering number estimate of $\mathcal{F}_{\text{NN}}(d_{\mathcal{Y}}, L, p, K, \kappa, M)$, which has been done in Lemma 6 and Lemma 7 in Liu et al. (2022). A direct consequence of these two lemmas is

$$T_2 \leq \frac{35 d_{\mathcal{Y}} L_{E_{\mathcal{Y}}^n}^2 R_{\mathcal{Y}}^2}{n} \log \mathcal{N}\left(\frac{\delta}{4 d_{\mathcal{Y}} L_{E_{\mathcal{Y}}^n}}, \mathcal{F}_{\text{NN}}, \|\cdot\|_{\infty}\right) + 6\delta$$

$$\lesssim \frac{d_{\mathcal{Y}}^2 K L_{\Phi}^2}{n}\left(\ln \delta^{-1} + \ln L + \ln(pB) + L\ln\kappa + L\ln p\right) + \delta$$

$$\lesssim \frac{d_{\mathcal{Y}}^2 K L_{\Phi}^2}{n}\left(\ln \delta^{-1} + \ln(B) + L\ln\kappa + L\ln p\right) + \delta,$$

where we used Lemma 6 and 7 from Liu et al. (2022) for the second inequality. The constant in $\lesssim$ depends on $L_{E_{\mathcal{Y}}^n}$ and $R_{\mathcal{X}}$. Substituting parameters $K, B, \kappa$ from equation 30, the above estimate gives

$$T_2 \lesssim L_{\Phi}^2 d_{\mathcal{Y}}^2 n^{-1} pL\left(\ln \delta^{-1} + L\ln B + L\ln R + L\ln p\right) + \delta$$

$$\lesssim L_{\Phi}^2 d_{\mathcal{Y}}^2 n^{-1} pL(\ln \delta^{-1} + L^2) + \delta$$

$$\lesssim L_{\Phi}^2 d_{\mathcal{Y}}^2 n^{-1} p\left(L^3 + (\ln \delta^{-1})^2\right) + \delta,$$

where we used the fact $\ln \delta^{-1} \lesssim L$ with the choice equation 34. The constant in $\lesssim$ depends on $L_{E_{\mathcal{Y}}^n}, L_{D_{\mathcal{Y}}^n}, L_{E_{\mathcal{X}}^n}, L_{D_{\mathcal{X}}^n}, R_{\mathcal{X}}$ and $d_{\mathcal{X}}$. We further substitute the values of $p$ and $L$ from equation 30 into the above estimate

$$T_2 \lesssim L_{\Phi}^{2-d_{\mathcal{X}}} d_{\mathcal{Y}}^3 n^{-1} \varepsilon_1^{-d_{\mathcal{X}}}\left(d_{\mathcal{X}}^3(\ln \varepsilon_1^{-1})^3 + (\ln \delta^{-1})^2\right) + \delta$$

Combining the $T_1$ estimate above and the $T_2$ estimate in equation 33 yields that

$$T_1 + T_2 \lesssim d_{\mathcal{Y}}\varepsilon_1^2 + L_{\Phi}^{2-d_{\mathcal{X}}} d_{\mathcal{Y}}^3 n^{-1} \varepsilon_1^{-d_{\mathcal{X}}}\left((\ln \varepsilon_1^{-1})^3 + (\ln \delta^{-1})^2\right)$$
$$+ L_{\Phi}^2 \mathbb{E}_u\left[\|\Pi_{\mathcal{X},d_{\mathcal{X}}}^n(u) - u\|_{\mathcal{X}}^2\right] + \sigma^2 + \delta.$$

In order to balance the above error, we choose

$$\delta = n^{-1}, \quad \varepsilon_1 = d_{\mathcal{Y}}^{\frac{2}{2+d_{\mathcal{X}}}} n^{-\frac{1}{2+d_{\mathcal{X}}}}. \tag{34}$$

Therefore,

$$T_1 + T_2 \lesssim d_{\mathcal{Y}}^{\frac{6+d_{\mathcal{X}}}{2+d_{\mathcal{X}}}} n^{-\frac{2}{2+d_{\mathcal{X}}}} (1 + L_{\Phi}^{2-d_{\mathcal{X}}})\left((\ln \frac{n}{d_{\mathcal{Y}}})^3 + (\ln n)^2\right)$$
$$+ L_{\Phi}^2 \mathbb{E}_u\left[\|\Pi_{\mathcal{X},d_{\mathcal{X}}}^n(u) - u\|_{\mathcal{X}}^2\right] + \sigma^2 + n^{-1}, \tag{35}$$

where we combine the choice in equation 34 and equation 30 as

$$L = \Omega(\ln(\frac{n}{d_{\mathcal{Y}}})), \quad p = \Omega(d_{\mathcal{Y}}^{\frac{2-d_{\mathcal{X}}}{2+d_{\mathcal{X}}}} n^{\frac{d_{\mathcal{X}}}{2+d_{\mathcal{X}}}}),$$
$$K = \Omega(pL), \quad \kappa = \Omega(M^2), \quad M \geq \sqrt{d_{\mathcal{Y}}} L_{E_{\mathcal{X}}^n} R_{\mathcal{Y}}.$$

Here the notation $\Omega$ contains constants that depends on $L_{E_{\mathcal{Y}}^n}, L_{D_{\mathcal{Y}}^n}, L_{E_{\mathcal{X}}^n}, L_{D_{\mathcal{X}}^n}, R_{\mathcal{X}}$ and $d_{\mathcal{X}}$.

Combining equation 25 and equation 35, we have

$$\mathbb{E}_{\mathcal{S}}\mathbb{E}_{u\sim\gamma}\left[\|D_{\mathcal{Y}}^n \circ \Gamma_{\text{NN}} \circ E_{\mathcal{X}}^n(u) - \Phi(u)\|_{\mathcal{Y}}^2\right] \lesssim d_{\mathcal{Y}}^{\frac{6+d_{\mathcal{X}}}{2+d_{\mathcal{X}}}} n^{-\frac{2}{2+d_{\mathcal{X}}}} (1 + L_{\Phi}^{2-d_{\mathcal{X}}}))\left((\ln \frac{n}{d_{\mathcal{Y}}})^3 + (\ln n)^2\right)$$
$$+ L_{\Phi}^2 \mathbb{E}_u\left[\|\Pi_{\mathcal{X},d_{\mathcal{X}}}^n(u) - u\|_{\mathcal{X}}^2\right] + \mathbb{E}_{\mathcal{S}}\mathbb{E}_{w\sim\Phi_{\#}\gamma}\left[\|\Pi_{\mathcal{Y},d_{\mathcal{Y}}}^n(w) - w\|_{\mathcal{Y}}^2\right]$$
$$+ \sigma^2 + n^{-1}.$$

$\square$

*Proof of Theorem 2.* Similarly to the proof of Theorem 1, we have

$$\mathbb{E}_{\mathcal{S}}\mathbb{E}_u\|D_{\mathcal{Y}}^n \circ \Gamma_{\mathrm{NN}} \circ E_{\mathcal{X}}^n(u) - \Phi(u)\|_{\mathcal{Y}}^2 \leq \mathrm{I} + \mathrm{II},$$

and

$$\mathrm{I} \leq 2L_{D_{\mathcal{Y}}^n}^2\left(T_1 + T_2\right),$$

where $T_1$ and $T_2$ are defined in equation 26. Following the same procedure in equation 27, we have

$$T_1 \leq 4\inf_{\Gamma \in \mathcal{F}_{\mathrm{NN}}} \mathbb{E}_u\left[\|\Gamma \circ E_{\mathcal{X}}^n(u) - E_{\mathcal{Y}}^n \circ \Phi(u)\|_2^2\right] + 6\mathbb{E}_{\mathcal{S}_2}\frac{1}{n}\sum_{i=n+1}^{2n}\|\varepsilon_i\|_{\mathcal{Y}}^2.$$

To obtain an approximation of the discretized target map $\Gamma_d^n := E_{\mathcal{Y}}^n \circ \Phi \circ D_{\mathcal{X}}^n$, we apply the following lemma for each component function of $\Gamma_d^n$.

**Lemma 3** (Theorem 1.1 in Shen et al. (2019)). *Given $f \in C([0,1]^d)$, for any $L \in \mathbb{N}^+$, $p \in \mathbb{N}^+$, there exists a function $\phi$ implemented by a ReLU FNN with width $3^{d+3}\max\{d\lfloor p^{1/d}\rfloor, p+1\}$, and depth $12L + 14 + 2d$ such that*

$$\|f - \phi\|_\infty \leq 19\sqrt{d}\omega_f(p^{-2/d}L^{-2/d}),$$

*where $\omega_f(\cdot)$ is the modulus of continuity.*

Apply Lemma 3 to each component $h_i$ of $\Gamma_d^n$, we can find a neural network $\tilde{h}_i \in \mathcal{F}_{\mathrm{NN}}(1, L, \tilde{p}, M)$ such that

$$\|h_i - \tilde{h}_i\|_\infty \leq CL_\Phi\varepsilon_1,$$

where $L, \tilde{p} > 0$ are integers such that $Lp = \lceil \varepsilon_1^{-d_{\mathcal{X}}/2}\rceil$, and the constant $C$ depends on $d_{\mathcal{X}}$. Assembling the neural networks $\tilde{h}_i$ together, we can find a neural network $\tilde{\Gamma}_d^n$ in $\mathcal{F}_{\mathrm{NN}}(d_{\mathcal{Y}}, L, p, M)$ with $p = d_{\mathcal{Y}}\tilde{p}$, such that

$$\|\tilde{\Gamma}_d^n - \Gamma_d^n\|_\infty \leq CL_\Phi\varepsilon_1.$$

Similarly to the derivations in equation equation 31 and equation 32, we obtain that

$$T_1 \lesssim L_\Phi^2 d_{\mathcal{Y}}\varepsilon_1^2 + L_\Phi^2\mathbb{E}_u\left[\|\Pi_{\mathcal{X},d_{\mathcal{X}}}^n(u) - u\|_{\mathcal{X}}^2\right] + \sigma^2, \tag{36}$$

where the notation $\lesssim$ contains constants that depend on $d_{\mathcal{X}}$ and $L_{E_{\mathcal{Y}}^n}$. To deal with term $T_2$, we apply the following lemma concerning the covering number.

**Lemma 4.** *[Lemma 10 in Liu et al. (2022)] Under the conditions of Theorem 2, we have*

$$T_2 \leq \frac{35d_{\mathcal{Y}}R_{\mathcal{Y}}^2}{n}\log\mathcal{N}\left(\frac{\delta}{4d_{\mathcal{Y}}L_{E_{\mathcal{Y}}^n}R_{\mathcal{Y}}}, \mathcal{F}_{\mathrm{NN}}, 2n\right) + 6\delta.$$

Combining Lemma 4 with equation 36, we derive that

$$\begin{aligned}\mathrm{I} \leq &CL_\Phi^2 L_{D_{\mathcal{Y}}^n}^2 d_{\mathcal{Y}}\varepsilon_1^2 + 16L_{D_{\mathcal{Y}}^n}^2 L_{E_{\mathcal{Y}}^n}^2 L_\Phi^2\mathbb{E}_u\left[\|\Pi_{\mathcal{X},d_{\mathcal{X}}}^n(u) - u\|_{\mathcal{X}}^2\right] + 12L_{D_{\mathcal{Y}}^n}^2\sigma^2 \\ &+ \frac{70L_{D_{\mathcal{Y}}^n}^2 d_{\mathcal{Y}}R_{\mathcal{Y}}^2}{n}\log\mathcal{N}\left(\frac{\delta}{4d_{\mathcal{Y}}L_{E_{\mathcal{Y}}^n}R_{\mathcal{Y}}}, \mathcal{F}_{\mathrm{NN}}(d_{\mathcal{Y}}, L, p, M), 2n\right) + 12L_{D_{\mathcal{Y}}^n}^2\delta.\end{aligned} \tag{37}$$

By the definition of covering number (c.f. Definition 5 in Liu et al. (2022)), we first note that the covering number of $\mathcal{F}_{\mathrm{NN}}(d_{\mathcal{Y}}, L, p, M)$ is bounded by that of $\mathcal{F}_{\mathrm{NN}}(1, L, p, M)$:

$$\mathcal{N}\left(\frac{\delta}{4d_{\mathcal{Y}}L_{E_{\mathcal{Y}}^n}R_{\mathcal{Y}}}, \mathcal{F}_{\mathrm{NN}}(d_{\mathcal{Y}}, L, p, M), 2n\right) \leq Ce^{d_{\mathcal{Y}}}\mathcal{N}\left(\frac{\delta}{4d_{\mathcal{Y}}L_{E_{\mathcal{Y}}^n}R_{\mathcal{Y}}}, \mathcal{F}_{\mathrm{NN}}(1, L, p, M), 2n\right).$$

Thus it suffices to find an estimate on the covering number of $\mathcal{F}_{\mathrm{NN}}(1, L, p, M)$. A generic bound for classes of functions is provided by the following lemma.

**Lemma 5** (Theorem 12.2 of Anthony et al. (1999)). *Let $F$ be a class of functions from some domain $\Omega$ to $[-M, M]$. Denote the pseudo-dimension of $F$ by $\mathrm{Pdim}(F)$. For any $\delta > 0$, we have*

$$\mathcal{N}(\delta, F, m) \leq \left( \frac{2eMm}{\delta \mathrm{Pdim}(F)} \right)^{\mathrm{Pdim}(F)} \tag{38}$$

*for $m > \mathrm{Pdim}(F)$.*

The next lemma shows that for a DNN $\mathcal{F}_{\mathrm{NN}}(1, L, p, M)$, its pseudo-dimension of can be bounded by the network parameters.

**Lemma 6** (Theorem 7 of Bartlett et al. (2019)). *For any network architecture $\mathcal{F}_{\mathrm{NN}}$ with $L$ layers and $U$ parameters, there exists an universal constant $C$ such that*

$$\mathrm{Pdim}(\mathcal{F}_{\mathrm{NN}}) \leq CLU \log(U). \tag{39}$$

For the network architecture $\mathcal{F}_{\mathrm{NN}}(1, L, p, M)$, the number of parameters is bounded by $U = Lp^2$. We apply Lemma 5 and 6 to bound the covering number by its parameters:

$$\log \mathcal{N} \left( \frac{\delta}{4 d_{\mathcal{Y}} L_{E_{\mathcal{Y}}^n} R_{\mathcal{Y}}}, \mathcal{F}_{\mathrm{NN}}(d_{\mathcal{Y}}, L, p, M), 2n \right) \leq C_1 d_{\mathcal{Y}} p^2 L^2 \log \left( p^2 L \right) \left( \log \left( \frac{R_{\mathcal{X}}^2 d_{\mathcal{Y}} L_{E_{\mathcal{Y}}^n} L_{\Phi}}{L^2 p^2 \log(Lp^2)} \right) + \log \delta^{-1} + \log n \right), \tag{40}$$

when $2n > C_2 p^2 L^2 \log(p^2 L)$ for some universal constants $C_1$ and $C_2$. Note that $p, L$ are integers such that $pL = \left\lceil d_{\mathcal{Y}} \varepsilon_1^{-d_{\mathcal{X}}/2} \right\rceil$, therefore we have

$$\log \mathcal{N} \left( \frac{\delta}{4 d_{\mathcal{Y}} L_{E_{\mathcal{Y}}^n} R_{\mathcal{Y}}}, \mathcal{F}_{\mathrm{NN}}(d_{\mathcal{Y}}, L, p, M), 2n \right) \lesssim d_{\mathcal{Y}}^3 \varepsilon_1^{-d_{\mathcal{X}}} \log(d_{\mathcal{Y}} \varepsilon_1^{-1}) \left( \log L_{\Phi} - \log(d_{\mathcal{Y}} \varepsilon^{-1}) + \log \delta^{-1} + \log n \right), \tag{41}$$

where the notation $\lesssim$ contains constants that depend on $R_{\mathcal{X}}, d_{\mathcal{X}}$ and $L_{E_{\mathcal{Y}}^n}$.

Substituting the above covering number estimate back to equation 37 gives

$$\mathrm{I} \lesssim L_{\Phi}^2 d_{\mathcal{Y}} \varepsilon_1^2 + L_{\Phi}^2 \mathbb{E}_u \left[ \| \Pi_{\mathcal{X}, d_{\mathcal{X}}}^n(u) - u \|_{\mathcal{X}}^2 \right] + \sigma^2$$
$$+ L_{\Phi}^2 n^{-1} d_{\mathcal{Y}}^4 \varepsilon_1^{-d_{\mathcal{X}}} \log(d_{\mathcal{Y}} \varepsilon_1^{-1}) \left( \log L_{\Phi} - \log(d_{\mathcal{Y}} \varepsilon^{-1}) + \log \delta^{-1} + \log n \right) + \delta,$$

where the notation $\lesssim$ contains constant that depends on $L_{E_{\mathcal{Y}}^n}, L_{D_{\mathcal{Y}}^n}, L_{E_{\mathcal{X}}^n}, L_{D_{\mathcal{X}}^n}, R_{\mathcal{X}}$ and $d_{\mathcal{X}}$. Letting

$$\varepsilon_1 = d_{\mathcal{Y}}^{\frac{3}{2+d_{\mathcal{X}}}} n^{-\frac{1}{2+d_{\mathcal{X}}}}, \delta = n^{-1},$$

we have

$$\mathrm{I} \lesssim L_{\Phi}^2 d_{\mathcal{Y}}^{\frac{8+d_{\mathcal{X}}}{2+d_{\mathcal{X}}}} n^{-\frac{2}{2+d_{\mathcal{X}}}} + L_{\Phi}^2 \mathbb{E}_u \left[ \| \Pi_{\mathcal{X}, d_{\mathcal{X}}}^n(u) - u \|_{\mathcal{X}}^2 \right] + (\sigma^2 + n^{-1}) + L_{\Phi}^2 \log(L_{\Phi}) d_{\mathcal{Y}}^{\frac{8+d_{\mathcal{X}}}{2+d_{\mathcal{X}}}} n^{-\frac{2}{2+d_{\mathcal{X}}}} \log(n)$$
$$\lesssim L_{\Phi}^2 \log(L_{\Phi}) d_{\mathcal{Y}}^{\frac{8+d_{\mathcal{X}}}{2+d_{\mathcal{X}}}} n^{-\frac{2}{2+d_{\mathcal{X}}}} \log n + (\sigma^2 + n^{-1}) + L_{\Phi}^2 \mathbb{E}_u \left[ \| \Pi_{\mathcal{X}, d_{\mathcal{X}}}^n(u) - u \|_{\mathcal{X}}^2 \right], \tag{42}$$

where $\lesssim$ contains constants that depend on $L_{E_{\mathcal{Y}}^n}, L_{D_{\mathcal{Y}}^n}, L_{E_{\mathcal{X}}^n}, L_{D_{\mathcal{X}}^n}, R_{\mathcal{X}}$ and $d_{\mathcal{X}}$. Combining our estimate equation 42 and equation 25, we have

$$\mathbb{E}_{\mathcal{S}} \mathbb{E}_u \| D_{\mathcal{Y}}^n \circ \Gamma_{\mathrm{NN}} \circ E_{\mathcal{X}}^n(u) - \Psi(u) \|_{\mathcal{Y}}^2 \lesssim L_{\Phi}^2 \log(L_{\Phi}) d_{\mathcal{Y}}^{\frac{8+d_{\mathcal{X}}}{2+d_{\mathcal{X}}}} n^{-\frac{2}{2+d_{\mathcal{X}}}} \log n + (\sigma^2 + n^{-1})$$
$$+ L_{\Phi}^2 \mathbb{E}_u \left[ \| \Pi_{\mathcal{X}, d_{\mathcal{X}}}^n(u) - u \|_{\mathcal{X}}^2 \right] + \mathbb{E}_{\mathcal{S}} \mathbb{E}_{w \sim \Phi_{\#} \gamma} \left[ \| \Pi_{\mathcal{Y}, d_{\mathcal{Y}}}^n(w) - w \|_{\mathcal{Y}}^2 \right].$$

$\square$

*Proof of Theorem 3.* Under Assumption 5, the target finite dimensional map becomes $\Gamma_d^n := E_{\mathcal{Y}} \circ \Phi \circ D_{\mathcal{X}} : \mathcal{M} \to \mathbb{R}^{d_{\mathcal{Y}}}$, which is a Lipschitz map defined on $\mathcal{M} \subset \mathbb{R}^{d_{\mathcal{X}}}$. Similar to the proof of Theorem 2, the generalization error is decomposed as the following

$$\mathbb{E}_{\mathcal{S}}\mathbb{E}_u \|D_{\mathcal{Y}} \circ \Gamma_{\mathrm{NN}} \circ E_{\mathcal{X}}(u) - \Phi(u)\|_{\mathcal{Y}}^2 \le T_1 + T_2 + \mathrm{II}, \tag{43}$$

where $T_1, T_2$ and $\mathrm{II}$ are defined in equation 26 and equation 25 respectively. Following the same procedure in equation 27, we obtained that

$$T_1 \le 4 \inf_{\Gamma \in \mathcal{F}_{\mathrm{NN}}} \mathbb{E}_u \left[\|\Gamma \circ E_{\mathcal{X}}^n(u) - E_{\mathcal{Y}}^n \circ \Phi(u)\|_2^2\right] + 6\mathbb{E}_{\mathcal{S}_2} \frac{1}{n} \sum_{i=n+1}^{2n} \|\varepsilon_i\|_{\mathcal{Y}}^2.$$

We then replace Lemma 3 by the following modified version of lemma 17 from Liu et al. (2022) to obtain an FNN approximation to $\Gamma_d^n$.

**Lemma 7** (Lemma 17 in Liu et al. (2022)). *Suppose assumption 5 holds, and assume that $\|a\|_\infty \le B$ for all $a \in \mathcal{M}$. For any Lipschitz function $f$ with Lipschitz constant $R$ on $\mathcal{M}$, and any integers $\tilde{L}, \tilde{p} > 0$, there exists $\tilde{f} \in \mathcal{F}_{\mathrm{NN}}(1, L, p, M)$ such that*

$$\|\tilde{f} - f\|_\infty \le CR\tilde{L}^{-\frac{2}{d_0}} \tilde{p}^{-\frac{2}{d_0}},$$

*where the constant $C$ solely depends on $d_0, B, \tau$ and the surface area of $\mathcal{M}$. The parameters of $\mathcal{F}_{\mathrm{NN}}(1, L, p, M)$ are chosen as the following*

$$L = \Omega(\tilde{L} \log \tilde{L}), p = \Omega(d_{\mathcal{X}}\tilde{p} \log \tilde{p}), M = R.$$

*The constants in $\Omega$ depend on $d_0, B, \tau$ and the surface area of $\mathcal{M}$.*

Apply the above lemma to each component of $E_{\mathcal{Y}} \circ \Phi \circ D_{\mathcal{X}}$ and assemble all individual neural networks together, we obtain a neural network $\tilde{\Gamma}_d^n \in F(d_{\mathcal{Y}}, L, p, M)$ such that

$$\|\tilde{\Gamma}_d^n - \Gamma_d^n\|_\infty \lesssim L_\Phi \varepsilon,$$

Here the parameters $L = \Omega(\tilde{L} \log \tilde{L})$, $p = \Omega(d_{\mathcal{X}} d_{\mathcal{Y}} \tilde{p} \log \tilde{p})$, $M = \Omega(L_\Phi)$ with $\tilde{L}\tilde{p} = \Omega(\varepsilon)$. The notation $\lesssim$ and $\Omega$ contains constants that solely depend on $d_0, R_{\mathcal{X}}, L_{E_{\mathcal{X}}}, \tau$ and surface area of $\mathcal{M}$. The rest of the proof follows the same procedure as in proof of Theorem 2.

$\square$

*Proof of Theorem 4.* The proof is similar to that of Theorem 2 with a slight change of the neural network construction, so we only provide a brief proof below.

While Assumption 6 holds, the target map $\Phi : \mathcal{X} \mapsto \mathcal{Y}$ can be decomposed as the following

$$\begin{array}{c}
\mathbb{R}^{d_0} \xrightarrow{g_1} \mathbb{R} \\
\nearrow^{V_1} \quad \searrow \\
\mathcal{X} \xrightarrow{E_{\mathcal{X}}^n} \mathbb{R}^{d_{\mathcal{X}}} \xrightarrow{V_i} \mathbb{R}^{d_0} \xrightarrow{g_i} \mathbb{R} \longrightarrow \mathbb{R}^{d_{\mathcal{Y}}} \xrightarrow{D_{\mathcal{Y}}^n} \mathcal{Y}. \\
\searrow_{V_{d_{\mathcal{Y}}}} \quad \nearrow \\
\mathbb{R}^{d_0} \xrightarrow{g_{d_{\mathcal{Y}}}} \mathbb{R}
\end{array} \tag{44}$$

Notice that each route contains a composition of a linear function $V_i$ and a nonlinear map $g_i : \mathbb{R}^{d_0} \to \mathbb{R}$. The nonlinear function $g_i$ can be approximated by a neural network with a size that is independent from $d_{\mathcal{X}}$, while the linear functions $V_i$ can be learned through a linear layer of neural network. Consequently, the function $h_i := V_i \circ g_i$ can be approximated by a neural network $\tilde{h}_i \in \mathcal{F}_{\mathrm{NN}}(1, L+1, \tilde{p}, M)$ such that

$$\|h_i - \tilde{h}_i\|_\infty \le CL_\Phi \varepsilon$$

where $L, \tilde{p} > 0$ are integers with $Lp = \lceil \varepsilon_1^{-d_0/2} \rceil$, and the constant $C$ depends on $d_0$. Assembling the neural networks $\tilde{h}_i$ together, we can find a neural network $\tilde{\Gamma}_d^n$ in $\mathcal{F}_{\mathrm{NN}}(d_{\mathcal{Y}}, L+1, p, M)$ with $p = d_{\mathcal{Y}}\tilde{p}$, such that

$$\|\tilde{\Gamma}_d^n - \Gamma_d^n\|_\infty \le CL_\Phi \varepsilon_1 \,.$$

The rest of the proof follows the same as in the proof of Theorem 2.

$\square$

*Proof of Theorem 5.* The proof is very similar to that of Theorem 4. Under Assumption 7, the target map $\Phi$ has the following structure:

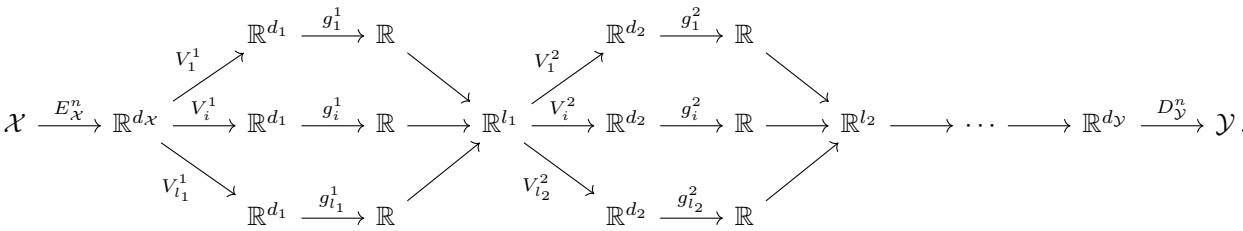

$$(45)$$

where the abbreviation notation $\cdots$ denotes blocks $G^i, i = 3, \ldots, G^k$. The neural network construction for each block $G^i$ is the same as in the proof of Theorem 4. Specifically, there exists a neural network $H_i \in \mathcal{F}_{\mathrm{NN}}(l_i, L+1, l_i\tilde{p}, M)$ such that

$$\|G^i - H^i\|_\infty \le CL_{G_i}\varepsilon_1 \,, \ \text{for all } i = 1, \ldots, k.$$

Concatenate all neural networks $H_i$ together, we obtain the following approximation

$$\|G^k \circ \cdots \circ G^1 - H^k \circ \cdots \circ H^1\| \le CL_\Phi \varepsilon_1 \,.$$

The rest of the proof follows the same as in the proof of Theorem 2.

$\square$

### A.2 Lipschitz constant of parameter to solution map for Parametric elliptic equation

The solution $u$ to equation 24 is unique for any given boundary condition $f$ so we can define the solution map:

$$S_a : f \in H^1 \mapsto u \in H^{3/2}.$$

To obtain an estimate of the Lipschitz constant of the parameter-to-solution map $\Phi$, we compute the Frechét derivative $DS_a[\delta]$ with respect to $a$ and derive an upper bound of the Lipschitz constant. It can be shown that the Frechét derivative is

$$DS_a[\delta] : f \mapsto v_\delta,$$

where $v_\delta$ satisfies the following equation

$$\begin{cases} -\mathrm{div}(a(x)\nabla_x v_\delta(x)) = \mathrm{div}(\delta\nabla u)\,, & \text{in } \Omega, \\ v_\delta = 0\,, & \text{on } \partial\Omega. \end{cases}$$

The above claim can be proved by using standard linearization argument and adjoint equation methods. Using classical elliptic regularity results, we derive that

$$\|v_\delta\|_{H^{3/2}} \le C\|\mathrm{div}(\delta\nabla u)\|_{H^{-1/2}}$$
$$\le C\|\delta\|_{L^\infty}\|u\|_{H^{3/2}} \le C\|\delta\|_{L^\infty}\|f\|_{H^1},$$

where $C$ solely depends on the ambient dimension $d = 2$ and $\alpha, \beta$. Therefore, the Lipschitz constant is $C\|f\|_{H^1}$.

