# OpenReview forum: "Deep Operator Learning Lessens the Curse of Dimensionality for PDEs"
_TMLR — Accepted by TMLR_

### Review · Reviewer_3b39 · 2023-06-17

**Summary Of Contributions:**

This paper focuses on providing theoretical evidence to explain why many PDE operators can be effectively approximated by deep neural networks (DNNs). In comparison to the existing work, this paper generalizes the setting from Hilbert space to Banach spaces that do not have an inner-product structure. Additionally, by assuming inherent low dimensionality and complexity in the target PDE operators (which may be embedded in a high dimensional space), they show that the sample complexity only depends on the low inherent dimensionality, which justifies why DNNs can mitigate the curse of dimensionality problem (CoD). Additional results and discussions are given on the discretization invariance neural networks.

**Audience:**

Yes

**Broader Impact Concerns:**

There aren't any major ethical implications of the work.

**Claims And Evidence:**

Yes

**Requested Changes:**

## Critical changes
1. As mentioned in weaknesses, there should be some discussion on why getting rid of the inner product structure is important. Besides, it will be helpful to also discuss how this relaxation can benefit the ML research (e.g., there are some problems that we cannot define inner products).

## Minors changes
1. p2, above Sec 1.1, 'To the best of our knowledge, the [typo] ... '
2. Above Assumption 1, 'We consider ... on the target PDE map [should be $\Phi$?], the encoder'
3. In proposition 1, $C$ and $h$ are not defined.

**Strengths And Weaknesses:**

## Strengths
1. This is an interesting topic to investigate as we have seen several works in ML (like consistency mode in diffusion) that need to implement  ODE/PDE solvers using neural networks (NNs). The work on this topic could provide some theoretical guidance in network design.
2. The analysis that explains how NNs alleviate the CoD in the PDE solving problems is interesting. Although this is not a very surprising result, I think it should be useful to present everything rigorously.
3. The paper is well-written and easy to follow for most of the part.

## Weaknesses
1. The authors claim that ``the removal of the inner-product assumption is crucial .. enabling us to apply the estimates to various PDE problems'',  and they mentioned in Sec 1.1 that they will give some demonstration in Section 3. However, it seems like the authors do not explicitly discuss this issue in Section 3.

---

> ### Author Response · Authors · 2023-06-19
> **Thank you for the review**
>
> Response:
> We thank the reviewer for their constructive review. We address the concerns and questions below.
>
> Q1&Critical changes: discussion on the removal of inner-product assumption
>
> A. Thanks for bringing this to our attention. We have added further discussions in the introduction section, and a minor description in section 3. All changes are marked in blue.
>
> Minor changes:
> We thank the reviewer for pointing this out. We have fixed all typos.

---

### Review · Reviewer_DUAy · 2023-07-05

**Summary Of Contributions:**

* This paper studies the sample complexity of learning an operator $\Phi$ from a Banach space $\mathcal{X}$ to a Banach space $\mathcal{Y}$ using a neural network.

* The procedure is to learn encoders and decoders $E_{\mathcal{X}} : \mathcal{X} \to \mathbb{R}^{d_X}$, $D_{\mathcal{X}} : \mathbb{R}^{d_X} \to \mathcal{X}$, $E_{\mathcal{Y}} : \mathcal{Y} \to \mathbb{R}^{d_Y}$, $D_{\mathcal{Y}} : \mathbb{R}^{d_Y} \to \mathcal{Y}$, as well as a neural-network transformation $\Gamma : \mathbb{R}^{d_X} \times \mathbb{R}^{d_Y}$. The neural network is chosen via empirical risk minimization (ERM) on the square loss.

* It is shown that ERM can avoid the curse of dimensionality if the operator to $\Phi$ has structure: it is a composition of low-dimensional functions.

* Several example applications to PDEs are given, where the spaces $\mathcal{X}$ and $\mathcal{Y}$ are function spaces that lack an inner product and hence previous theorems do not apply.

**Audience:**

Yes

**Claims And Evidence:**

Yes

**Requested Changes:**

### A) Comments about statements of theorems: [critical]

I had some difficulty understanding the statements of the theorems at a formal level. However, I am confident that the authors can clarify my understanding by answering my questions below.

1. In Theorems 1 and 3 what does it mean to solve the optimization problem for L = O(...), p = O(...) etc? Does it mean A, B, or C below?

* Possibility A) there are constants C_1,C_2, etc... depending on L_{E_Y^n}, etc... such that if we solve the optimization problem with $L = \lfloor{C_1 ln(n/d_Y)}\rfloor$, etc... we get the bound

* Possibility B) there are constants C_1,C_2, etc... depending on L_{E_Y^n}, etc... such that if we solve the optimization problem over all networks with $L <= \lfloor{C_1 ln(n/d_Y)} \rfloor$, p <= C_2 (...) etc.. we get the bound

* Possibility C) something else?

2. Another nit-pick is that in Theorem 3 there is reference to integers $\tilde{L}$, $\tilde{p}$ such that $\tilde{L}\tilde{p} = \lceil d_Y^{-3{d_0}/(4+2d_0)} n^{d_0 / (4+2d_0)} \rceil$. Should I interpret the theorem as applying to the optimization problem for any pair of integers satisfying this? Also, technically speaking what if $\lceil d_Y^{-3{d_0}/(4+2d_0)} n^{d_0 / (4+2d_0)} \rceil$ is prime, and either $\tilde{L}$ or $\tilde{p}$ has to be 1? (I have a similar comment about Theorem 2, equation (5), and Theorems 4 and 5.)

3.  Does Thm 3 also depend on Assumption 5? If not, why does $\sigma$ appear in the bound?

### B) Comments about the theorems: [critical]

I also have some questions about qualitatively interpreting the theorems. To summarize what confuses me: the theorems are stated as applying for a very specific choice of hyperparameters (width, depth, sparsity, weight bounds) without much explanation for that choice. I don't know how much of that choice of hyperparameters is arbitrary, and how much is flexible.

For example:

1. In Theorem 1, is there something special about d_X = 2? It seems that for d_X < 2, the width p must tend to zero as d_Y tends to infinity? But for d_X > 2, the width p can tend to infinity as d_Y tends to infinity? Why should there be such a behavior at d_X = 2? In comparison, in Theorem 2 there seems to be some special behavior at d_X = 4?

Is there anything going on here, or is the "interesting" regime of the theorems only for large d_X?

2. It would help the reader if the authors pointed out exactly what is different about Theorem 1 and Theorem 2. I don't understand how the sparsity and parameter L_{infinity} norm bounds are mattering. For example, why is M allowed to grow as sqrt(d_Y) in Theorem 2 but not in Theorem 1?

Similar questions about the hyperparameter choices in Theorems 3,4,5.

3. The bounds in the theorems are challenging to read. One suggestion to streamline the theorems and make them significantly easier to read and compare to each other would be to define something along the lines of
$\Pi_{err} = \sigma^2 + n^{-1} + L_{\Phi}^2 E_u[||\Pi_{X,d_X}^n(u) - u||^2_X] + E_S[E_{w\sim \Phi_{\\#} \gamma}[||\Pi_{Y,d_Y}^n(w) - w||^2_Y]$
, and then add $\Pi_{err}$ to each of theorems 1,2,3,4,5, since it is a shared term between each of these theorems. Similarly you could define something like
$RISK(\Gamma_{NN}) = E_S[E_u[||D_Y^n \circ \Gamma_{NN} \circ E_X^n(u) - \Phi(u)||^2_Y]]$
and reuse $RISK(\Gamma_{NN})$ in each of the theorems.

### C) Requested clarifications on discretization section [not critical]

1. What does it mean for all grid points x^i of all given discretized data to be denser than \hat{x}? Do you mean that the \hat{x} discretization is a subset of the x^i discretization?

### D) Comments on "Explicit complexity bounds for various PDE operator learning" [not critical]

1. In Section 3.1, it may help to write more explicitly with an equation how Assumption 7 is met. It took me some effort to understand this.

2. What does the "manifold dimension of the data set of media function a(x)" mean in Section 3.5? Is it always <=2 because Omega \subset R^2?

### Typos/very minor: [not critical]
* “discretization-invariant in operator learning”
* “are Banach space” -> "spaces"
* Overloaded \phi notation
* “and provides explicit” -> "and provide explicit"
* "hold ture"
* In Theorems 4,5 there is no O(\cdot) notation, but it's referred to in the statement.
* In discussion after Theorem 3, please clarify what "algebraic dependency" means. Does it mean that the bounds depend at most polynomially?
* Bottom of page 7: "where the sampling locations $x^i = [x_1^1,...,x_{s_i}^i]$
* Top of page 8: mismatch between r^i and r_i
* Missing unclosed parenthesis in (15)
* "is give by" -> "is given by"
* Equation (20), should have $\lesssim$ instead of $\leq$
* "Assumption 7 hold" -> "Assumption 7 holds"
* "see Appendix A.2" -> "see Appendix A.2."
* On page 3, the set $\\mathcal{S}_1$ is referred to a training set, and $\mathcal{S}_2$ as a test set. However, judging from the proof in the appendix it seems that $\\mathcal{S}1$ is used to create the encoders/decoders, and $\\mathcal{S}2$ is used to learn $\\Gamma$. Am I misinterpreting this?

**Strengths And Weaknesses:**

Strengths: The paper is well-structured, the math seems correct, and the results are of interest to the neural networks/PDEs community. The example applications in PDEs are compelling.

Weaknesses: From a technical point of view, the proofs of the main theorems are small modifications of Liu et al. (2022), which provides similar results but when $\mathcal{X}$ and $\mathcal{Y}$ are Hilbert spaces (have inner product structure).

The statements of some of the theorems are difficult to read in the current version (see my comments below).

---

> ### Author Response · Authors · 2023-07-17
> **thank you for the review**
>
> Thank you for the insightful suggestions and comments.
>
> A.1 Reply: All the bounds for neural network parameters are lower bounds, that is, the big O notation in theorem 1 and 3 means there are constants $C_1$, $C_2$, etc… depending on $L_{E_Y^n}$, etc... such that if we solve the optimization problem over all networks with $L \geq C_1 (…)$, $p \geq C_2 (...) $ etc.. we get the bound. We have added a sentence to explain this in the notation section.
>
> A.2 Reply: We are sorry for the confusing statement. This is a lower bound estimate, so as long as $\tilde{L}\tilde{p}$ are lower bounded by $\left \lceil d_Y^{\frac{-3d_0}{4+2d_0}} n^{\frac{d_0}{4+2d_0}} \right\rceil$, then the conclusion of theorem 3 holds. We have changed the equal sign to greater or equal sign in all main theorems to avoid confusion.
>
> A.3 Reply: Yes, Theorem 3 also depends on Assumption 5. This is a typo and we have fixed it.
>
> B.1 Reply:  We thank the referee for bringing this question to our attention. The special case $d_X = 2$ results from the technical proof. Specifically, the construction of the optimal neural network requires the width $p$ scales at the order of $d_Y \epsilon^{-d_X}$, see (30). To balance the approximation error (term $T_1$) and the generalization error (term $T_2$), we choose $\varepsilon = d_Y^{2/(2+d_X)}$, this implies that width p scales as $d_Y^{(2-d_X)/(2+d_X)}$.
>
> Because the bounds for neural network parameters are lower bound estimates, when $d_X \geq $2, theorem 1 holds if $p$ is greater than a negative power of $d_Y$, that is, the width of the (best) neural network does not have to be large in terms of the output dimension $d_Y$ of the target map. If $d_X < 2$, the neural network width needs to be larger than $\sqrt{d_Y}$.
>
> Q: Is there anything going on here, or is the "interesting" regime of the theorems only for large $d_X$?
>
> Reply: our results implies that for $d_X \geq 2$, the neural network width does not need to be scale as the output dimension $d_Y$ of the target map increases. We have added one sentence on this  qualitative interpretation in Remark 3.
>
> B.2 Reply: We are sorry for the typo in the bound of M in theorem 1. Indeed, both theorem 1 and theorem 2 used similar arguments to bound error term $T_2$, thus requiring the same upper bound of M. We have updated this in theorem 1 and its proof. The main difference between theorem 1 and theorem 2 lies in the choice of neural network class (3) and (4) respectively. As a consequence, different constructions of optimal neural networks are employed in the proof of theorem 1 and theorem 2, see lemma 2 and lemma 4, which leads to different asymptotic lower bounds for $p$. In fact, Theorem 2 considers a more general neural network class, thus having an asymptotic bound $n^{1/2}$ in the large $d_X$ regime, whereas in theorem 1, $p$ is lower bounded by $n$. We have added further discussions in remark 3 to point out this difference.
>
> Q: Similar questions about the hyperparameter choices in Theorems 3,4,5.
>
> Reply: In theorem 3,4,5, similar arguments hold to bound the error term $T_2$ so they all require the same bounds for parameter $M$. As for the lower bound of width p, they all have the same asymptotic bound (negative power of $d_Y$ times $n^{1/2}$ when $d_X$ is large. For small $d_X$, different constructions of optimal neural network may lead to a different special value of $d_X$.
>
> B.3 Reply: we thank the referee for the suggestions for better readability. We have added the definitions of the generalization error, and a quantity for the projections errors and noises in section 2.2 and replaced these terms in all main results.
>
> C.1 Reply: this is a typo. We actually need to assume to $\hat{x}$ is denser than all grid points $x^i$, that is, each $x^i $ discretization is a subset of the discretization of $\hat{x}$. This is a necessary condition to obtain a uniform projection error bound in proposition 1. We have rephrased this sentence in the manuscript.
>
> D.1 Reply: An easier way to see this is that Assumption 7 holds for any linear map and the solution map in section 3.1, as a convolution, is a linear map. We have added more discussions below assumption 7 and in section 3.1 to address this.
>
> D.2 Reply: The data set of media function $a(x)$ contains vectors that are discretizations of 2D pictures and the size of the vector is the number of pixels for these 2D pictures. The Shepp-Logan data set are benchmark example for modeling slices of human chest/lungs images, which are modeled by several ellipsoids. Therefore, all these pictures live on a smooth $d_0$-manifold because they are smoothly depending on several parameters, for example, the center, axis length, and rotation angle of the ellipsoids. Here $d_0$ is typically 6-20, depending on how many ellipsoids are considered.
>
> We thank the referee for pointing out these mistakes. We have fixed all the typos and listed other minor issues below.
>
> To be continued in another comment

---

> > ### Author Response · Authors · 2023-07-17
> > **continued comment for minor issues**
> >
> > [continued from previous comment]
> >
> > Q: In discussion after Theorem 3, please clarify what "algebraic dependency" means. Does it mean that the bounds depend at most polynomially?
> >
> > Reply: Yes, it means the estimate depends at most polynomially on $d_X$ and $d_Y$. We have clarified this in the manuscript.
> >
> > Q: On page 3, the set $\mathcal{S}_1$ is referred to a training set, and $\mathcal{S}_2$ as a test set. However, judging from the proof in the appendix it seems that $\mathcal{S}_1$ is used to create the encoders/decoders, and $\mathcal{S}_2$ is used to learn the FNN. Am I misinterpreting this?
> >
> > Reply: We are sorry for the mistake. Here $S_1$ is the training data to learn the encoders/decoders and $S_2$ is the training data to learn FNN. We have changed the definition in section 2.1.

---

> > > ### Comment · Reviewer_DUAy · 2023-07-17
> > > **Response**
> > >
> > > Thanks for your response and clarifications. I looked over your changes, and they seem good to me. I am happy to recommend acceptance.

---

### Review · Reviewer_SeaA · 2023-07-09

**Summary Of Contributions:**

This paper presents an analysis to explain how using deep neural networks to learn a class of PDEs have demonstrated success in the field. in particular, the merits of DNN in reducing the sample/computational complexity for a class of PDEs whose operator functions are Lipschitz continuity and exhibits low-dimensional structure.

**Audience:**

Yes

**Broader Impact Concerns:**

I do not see any concerns in terms of ethical implication of this work, and authors acknowledges the limitation of this work in the end of the paper.

**Claims And Evidence:**

Yes

**Requested Changes:**

Please see the "weakness" in the above section.

**Strengths And Weaknesses:**

The strength of the paper lies on the proposal of the analysis, which is quite comprehensive and inspirational as I see.

 This paper proposed a few major theorems which explains the claimed "lessening the curse of dimensionality". Authors layouts the phenomenon and organizes the analysis very well, with very intuitive illustration and very detailed explanations. Also, the examples to demo the prevalence of "low-dimensional structure" and/or low sample complexity in many cases are very well written, and it serves as strong evidence to support the paper's main results.

There are limitation of this work and authors also stated some of them in the "limitation" section. While a few good examples were presented to demonstrate the generalization and common applicability of the theories, there are somes questions/concerns.

$\mathbf{1}.$ The Lipschitz constant does have a big impact in general for the difficulty of training deep neural networks. Are authors aware of cases where, given a very large constant, some deep neural networks of solving PDEs are very difficult to train? Or, instead, one could bypass such an issue by construction of a smaller Lipschitz constant to learn PDE operators.

$\mathbf{2}.$ It might not be necessary to conduct some empirical study in this paper. But, since this works targets at analyzing DNN (for PDEs), would it be better off conducting some experiments to support the claim in practice?

$\mathbf{3}.$ A trivial item:
Notation could be improved with some minor editing, e.g., the composition function symbols on page 3; also it would be better to add more explanations between theorems (or to explain each terms or some of them one by one to help audience gain better understanding).

---

> ### Author Response · Authors · 2023-07-17
> **thank you for the review**
>
> Thank you for the insightful suggestions and comments.
>
> Q1 Reply:
> This is an interesting question that we are not able to fully answer in the current paper. One PDE example is the solution map of the Burgers equation with large terminal time T, where a shock (or a large derivative) may appear, thus the solution map is no longer a Lipschitz map. While the shock solution seems to have a simple structure in 1D, our theory may not apply due to the simple FNN architecture. In this case, a more complicated encoder, or a different NN architecture, or augmented training data may be needed to bypass the difficulty of the unbounded Lipschitz constant. One related work toward this question is [1].
>
> $[1]$ Chen, Zhen, et al. "Deep neural network modeling of unknown partial differential equations in nodal space." Journal of Computational Physics 449 (2022): 110782.
>
> Q2 Reply: We did not add numerical experiments to support the claim for the following reasons:
>
> 1. The lessening of CoD phenomenon in learning PDE operators has been observed by many numerical works with various neural network architectures, for example, the FNO, DeepONet, and discretization-invariant neural networks like IAE-net. In particular, it is numerically observed that one does not need more training data to approximate a PDE map when the media/solutions of the PDE is of high resolutions. The number of training data needed only depends on the parameter $d_0$, which is PDE-dependent and resolution-independent, as pointed out by the paper.
>
> 2. In practice, it is hard to conduct numerical experiments to support the generalization error analysis in the current work or any other theoretical paper in a rigorous way. This is because all generalization theories are built for the global minimizer to bypass the optimization error, but in practice, we cannot guarantee that the global minimizer $\Gamma_\text{NN}$ can be obtained due the the non-convexity of the training loss.
>
> Q3 Reply:
> We thank the referee for the suggestions. We have rewritten the composition functions in (2) and add more explanations in the beginning of section 2.2. We have also rewritten all main theorems for better readability.

---

### Comment · Action_Editors · 2023-08-14
**Some further questions**

While we are still waiting for one of the final recommendations, I thought it would be quite useful if the authors could quickly clarify the following issues:

(1) As pointed out by Reviewer 3b39, the significance of removing the inner product structure is not well-articulated. In the revision, the authors added more of such claims but did not provide much insight on the technical challenges to extend to Banach spaces. A casual inspection of the proof seems to suggest that there is little, as the majority of the work was performed in the embedded finite dimensional space (equipped even with the Euclidean norm). Other than defining the Lipschitz constants wrt an abstract norm and applying Lemma 1, can the authors clarify what challenges were addressed in order to extend to a Banach space?

(2) It would be great if the authors could compare the NN approach with other more standard techniques in learning a PDE operator. Is there any advantage in terms of the generalization bounds? This discussion could put the main results of this work into proper context.

(3) Assumption 5 requires some justification, as it is unlikely to hold in practice (unless the encoder is linear). In fact, do the authors really need it in the proof? It'd be great if the authors explicitly point out where each assumption (1-5) was applied.

Minor comments:
(4) Dx circ Ex approx I_X (not I_{R^{d_X}}) towards the end of page 3 (similarly for D_y circ E_y)

(5) first paragraph of Section 2.2, the notation Pi_{X, d_X}^n appeared before its definition in Theorem 1.

(6) Explain what is the reach parameter tau in Assumption 6, and how does it affect learning

(7) Perhaps emphasize that the operator Phi is nonlinear throughout (as often it could be mistaken as a linear operator)

---

> ### Author Response · Authors · 2023-08-17
> **thank you for the suggestions**
>
> We thank the editor for carefully reading our revised manuscript and their helpful suggestions.
>
> (1). We list below a comparison of contributions and technical issues between Liu et al and our works:
> - We want to point out the main goal of our work and Liu et al is different. Liu et al considers general Lipschitz operators on Hilbert spaces, whereas our work focus on Lipschitz PDE operators on Banach spaces, and we hope our work can be helpful for the PDE operator learning community. We analyze a number of PDE examples and provide a ready-to-use theorems for learning PDE operators while there is no discussions on PDE operators in Liu et al.
> - Without the inner-product in Banach spaces, our work addressed the following challenges: 1. The estimate for term T_1 in Liu et al no longer holds, a new estimate for T_1 is derived in our result, which leads to different noise term in the main results. 2. As the embeddings in the finite dimensional space only preserves distance but not the inner-product, we derived the embedding errors in proposition 1 for discretization invariant neural networks, and in lemma 1 for embeddings into L^p spaces. 3. We derive Lipschitz constants for several PDEs, which provides explicit generalization error for PDE operator learning.
>
> We also want to point out that although part of the work is performed in the embedded finite dimensional space with Euclidean norm, the analysis also applies for the finite dimensional space with L^p norm because 1. the main tool is Lemma 2 (Yarotsky 2017) and lemma 3 (Shen et al). 2. we carefully make sure that no inner-product is used
>
> Some aspects of the discussion above have been added in the previous revision, we have also added more discussions in remark 3 about the technical challenge.
>
> (2). We thank the editor for bringing this question to us. We have added more reference on other PDE operator learning methods, including GAN’s variants and other generative models in the introduction. However, to the best of our knowledge, there are no theoretical results on generalization bounds of these methods for PDE operator learning, thus we are unable to compare them with our results.
>
> (3). We thank the referee for bringing this question. After checking the proof, we did not apply assumption 5 because no linear terms of the noise are obtained in the proof. This term was originally used in the analysis of Liu et al to remove the noise dependence with the inner-product structure.
> We have deleted assumption 5 and added reference explicitly where other assumptions are applied.
> Assumption 1,2 -> below (27)
> Assumption 3 -> below (25)
> Assumption 4 -> above (33)
>
> Minor comments:
> (4) Thanks. We have fixed this typo.
>
> (5) We are not able to find the notation before its definition.
>
> (6) We have added one sentence above theorem 3.
>
> (7) Thanks. we have added a few words in the introduction to emphasize that the operator is nonlinear.

---

### Decision · Action_Editors · 2023-08-28

**Recommendation:** Accept with minor revision

**Comment:**

This work continued the theoretical analysis of deep operator learning, where an encoder-decoder structure is first applied and then deep neural networks are employed to learn the resulting (lower dimensional) map. The results bear some similarity to those of Liu et al. (2022) and the main contribution lies on the analysis of several nonlinear PDE operators. All reviewers found this work well-written and recommended acceptance. The presented analysis and examples are interesting and may inspire further interaction between deep learning and PDEs.

One revision we would like to suggest is to compare the authors' results with those of Liu et al. (2022) in much greater detail, under the Hilbert setting. It seems the lack of an inner product (in this work) was circumvented by a simple application of the triangle inequality, which probably will make the authors' result a bit looser (c.f., Remark 3). If this is the case, it is important to explicitly acknowledge this trade-off (looser but more general) so that the tighter results of Liu et al. (2022) will not be shadowed by the authors' seemingly more general result (for any Banach space). It will also explain why the current work could afford to remove Assumption 5, even under a more general setting. Lastly, given how the proof proceeds, it is probably wise to not over-claim the extension to a Banach space. [Of course, this is not to say the case studies in Section 3 are not interesting or important.]

Notation: first paragraph of Section 2, do the authors mean x <= Cy for the notation x = O(y)? [This would be the usual meaning of big-O. The current wording seems to suggest the converse, which should be denoted as big-Omega.]

**Audience:**

Yes, for those who interested in applying deep learning to PDEs and those who apply PDE techniques to ML.

**Claims And Evidence:**

The analysis and proof seems to be correct and convincing. There are no numerical experiments involved.

---

> ### Author Response · Authors · 2023-09-22
> **further clarification on the inner-product**
>
> We thank the editor for the suggestions. We have marked all changes in blue for the convenience of the editor. They will be removed in the camera-ready version.
>
> We have edited the contribution to emphasize more on this tradeoff of accuracy for generalization. We also added one sentence in remark 3, and proof of theorem 1 to explicitly point out that we use triangle inequality to circumvent the inner-product structure.
>
> We agree with the editor that Assumption 5 is not needed in our current work. Specifically, in Liu et al, a linear term of the noise perturbation will show up in the inner-product when FOIL out quadratic terms. Assumption 5 is used to eliminate the linear terms, leading to a better noise decay in the final results. In our work, this is not possible because no inner product is allowed so Assumption 5 is no longer useful.
>
> Notation: to be consistent with the convention notation, we have changed it to big-\Omega notations. We also added the big-O notation as it was involved in some discussions of the paper. We have marked all changes in blue that involved big-Omega notation.